# HMGA2 directly mediates chromatin condensation in association with neuronal fate regulation

Naohiro Kuwayama[1,6], Tomoya Kujirai[2], Yusuke Kishi [2], Rina Hirano [2], Kenta Echigoya [2], Lingyan Fang [1], Sugiko Watanabe[3], Mitsuyoshi Nakao [3], Yutaka Suzuki [4], Kei-ichiro Ishiguro [3], Hitoshi Kurumizaka [2] ✉ & Yukiko Gotoh [1,5] ✉

Identification of factors that regulate chromatin condensation is important for understanding of gene regulation. High-mobility group AT-hook (HMGA) proteins 1 and 2 are abundant nonhistone chromatin proteins that play a role in many biological processes including tissue stem-progenitor cell regulation, but the nature of their protein function remains unclear. Here we show that HMGA2 mediates direct condensation of polynucleosomes and forms droplets with nucleosomes. Consistently, most endogenous HMGA2 localized to transposase 5– and DNase I–inaccessible chromatin regions, and its binding was mostly associated with gene repression, in mouse embryonic neocortical cells. The AT-hook 1 domain was necessary for chromatin condensation by HMGA2 in vitro and in cellulo, and an HMGA2 mutant lacking this domain was defective in the ability to maintain neuronal progenitors in vivo. Intrinsically disordered regions of other proteins could substitute for the AT-hook 1 domain in promoting this biological function of HMGA2. Taken together, HMGA2 may regulate neural cell fate by its chromatin condensation activity.

In eukaryotes, genomic DNA is compacted with histone proteins as chromatin, in which the nucleosome composed of four histones H2A, H2B, H3, and H4 is an elemental structure. Chromatin architecture dependent on the condensation state of the nucleosomes is an essential determinant of gene regulation. In addition to histones, nonhistone chromatin factors such as heterochromatin protein 1 (HP1), methyl-CpG binding protein 2 (MeCP2), and Polycomb group (PcG) proteins play pivotal roles in regulation of direct chromatin condensation. Identification of additional key regulators of chromatin condensation will be important to provide a better understanding of the intricate mechanisms underlying the control of gene expression.

High-mobility group AT-hook protein 1 (HMGA1) and HMGA2 are abundant nonhistone chromatin factors. Since their original identification as small proteins in a chromatin fraction[1], HMGA proteins have been implicated in the regulation of many biological processes including embryonic development (determination of body or tissue size), stem cell maintenance, cellular senescence, and tumorigenesis[2]. Knockout of both *Hmga1* and *Hmga2* in mice gives rise to a pygmy phenotype[3]. HMGA proteins have been found to regulate the proliferation and differentiation of various tissue stem-progenitor cells including mesenchymal, hematopoietic, muscle, and neural stem cells[4–7]. In the central nervous system, HMGA2 promotes the

[1]Graduate School of Pharmaceutical Sciences, The University of Tokyo, Tokyo 113-0033, Japan. [2]Institute for Quantitative Biosciences, The University of Tokyo, Tokyo 113-0032, Japan. [3]Institute of Molecular Embryology and Genetics, Kumamoto University, Kumamoto 860-0811, Japan. [4]Department of Computational Biology, Graduate School of Frontier Sciences, The University of Tokyo, Chiba 277-8561, Japan. [5]International Research Center for Neurointelligence (WPI-IRCN), The University of Tokyo, Tokyo 113-0033, Japan. [6]Present address: Department of Molecular and Cell Biology, University of California, Berkeley, CA, USA. ✉e-mail: kurumizaka@iqb.u-tokyo.ac.jp; ygotoh@mol.f.u-tokyo.ac.jp

proliferation of neural progenitor cells (NPCs) and confers neurogenic potential on these cells[6,8]. However, the molecular basis of these biological functions of HMGA proteins have remained unclear.

The proposed role of HMGA proteins in chromatin structure and gene expression has been controversial. Historically, HMGA proteins have been described mainly as factors that activate gene expression. For example, HMGA has been shown to activate expression of the interleukin-2 receptor α chain (*IL-2Rα*), interferon-β (*IFN-β*), *cyclin A2*, *PLAG1*, and *IGF2BP2* genes by direct binding to their regulatory elements[9–17]. In the case of the *IL-2Rα* and *IFN-β* gene loci, HMGA1 reportedly mediates formation of an enhanceosome complex that consists of several transcription factors including NF-κB, SRF, GATA, STAT5, and ELF1 at the *IL-2Rα* gene and NF-κB, ATF2, c-Jun, and IRF at the *IFN-β* gene[9–13]. HMGA1 has also been found to compete with the linker histone H1 at the chicken *β-globin* gene locus. H1 binds to the nucleosome and compacts the higher-order chromatin structure. Given that H1 mediates chromatin condensation, HMGA1 has been proposed to mediate "decondensation" of chromatin structure by competing with H1 binding for the nucleosome, resulting in transcriptional activation of the *β-globin* gene[18,19]. This competition model was supported by fluorescence recovery after photobleaching (FRAP) analysis, which revealed an increased mobility of green fluorescent protein (GFP)-tagged H1 in mouse fibroblasts after injection of HMGA1[20]. Recent studies have also implicated HMGA proteins in DNA repair and associated activation of gene expression, with HMGA2 having been shown to recruit histone variant γH2AX, resulting in DNA demethylation and gene activation at target loci[21].

Other studies have suggested that HMGA proteins associate with condensed chromatin and formation of senescence-associated heterochromatin foci. Immunostaining of HMGA1 and G-banding with Wright's stain revealed that HMGA1 localizes to dark band regions that may correspond to heterochromatin[22]. HMGA1 was also detected in a sonication-resistant nuclear fraction and knockdown of HMGA1 showed global downregulation of H3K9me3 in human fibroblast[23,24]. A genome-wide analysis of biotin-tagged HMGA1 or HMGA2 overexpressed in mouse embryonic stem (ES) cells also suggested that both proteins localize to condensed chromatin regions[25]. In senescent cells, overexpression of GFP-tagged HMGA1 or HMGA2 induced the formation of senescence-associated heterochromatic foci[26,27]. In the context of tumorigenesis, HMGA2 has been shown to repress a set of tumor suppressor genes such as those for CDH1 and BRCA1 by direct binding to their regulatory elements[28,29].

HMGA proteins have thus been implicated in both gene activation (associated with chromatin decondensation) and gene repression (associated with chromatin condensation). However, these opposite effects may not necessarily be direct. For example, an effect of HMGA expression on cell fate or cell state may result in indirect changes in chromatin structure and gene expression at loci not directly targeted by HMGA proteins. It is therefore important to investigate direct functions of HMGA proteins in regulation of chromatin with approaches such as biochemical reconstitution. Another possible pitfall of previous studies is their reliance on overexpression of HMGA proteins in cell lines. It is thus also important to investigate the biological functions and targets of endogenous HMGA proteins in vivo.

In the present study, we have investigated the functions of HMGA2 directly by in vitro reconstitution analysis with recombinant proteins as well as by cellular and biochemical analysis of endogenous HMGA2 in vivo. We found that HMGA2 directly mediates chromatin compaction. Furthermore, we identified the domain of HMGA2 responsible for this action and revealed that this domain is essential for the biological function of HMGA2 in the regulation of NPC fate in the developing mouse neocortex.

## Results

### HMGA2 forms a complex with histone H1

We examined the relation between HMGA2 and H1 in order to provide insight into the role of HMGA proteins in chromatin structure, given the central role of H1 in internucleosome compaction[30] and the apparent competition between HMGA proteins and H1 in an artificial system and in cell lines[19,20]. We therefore determined whether endogenous HMGA2 colocalizes with H1 in the embryonic mouse neocortex, in which HMGA2 has been shown to promote the neuronal fate and proliferation of NPCs[6,8]. Immunoprecipitation followed by mass spectrometry (IP-MS) analysis of HMGA2 in a chromatin fraction of cells isolated from the mouse neocortex at embryonic day (E) 11.5 revealed the association of HMGA2 with the somatic linker histone variants H1.1, H1.2, H1.3, H1.4, and H1.5 (Fig. 1a). This IP-MS analysis also revealed an association of HMGA2 with H2B and the DNA damage response factor MDC1 (mediator of DNA damage checkpoint protein 1) (Fig. 1a), with the latter association being consistent with the previously demonstrated interaction of HMGA2 with the DNA damage repair pathway[31]. We also performed IP-MS analysis with Neuro2A cells (mouse neuroblasts) overexpressing mouse HMGA2, and again detected an association of HMGA2 with H1.1, H1.2, H1.4, and H1.5 as well as with H2A, H2B, and MDC1 (Supplementary Fig. 1a). Together, these results suggest that HMGA2 forms a complex with H1 in cells.

We next tested whether HMGA2 interacts with H1 directly or indirectly in an in vitro reconstitution system with recombinant proteins. We employed human H1.2 as a representative somatic linker histone, and the nucleosome was reconstituted with 193 bp of the Widom 601 DNA sequence resulting the nucleosome containing 24 bp linker DNA at both ends. Pull-down of hexahistidine (His)-tagged human HMGA2 by Ni-nitrilotriacetic acid (Ni-NTA) beads did not result in the coprecipitation of H1.2 (Fig. 1b, lane 9), indicating that HMGA2 does not directly bind to H1.2. In contrast, the nucleosomal core histones were pulled down by His-tagged HMGA2 (Fig. 1b, lane 8), indicating direct binding of HMGA2 to the nucleosome. Interestingly, H1.2 was effectively coprecipitated with His-tagged HMGA2 in the presence of the nucleosome (Fig. 1b, lane 10). The coprecipitation of the nucleosome and H1.2 was not observed when the His tag was enzymatically removed from HMGA2 by the PreScission protease (Fig. 1b, lanes 6 and 7). An electrophoretic mobility shift assay (EMSA) confirmed that HMGA2 alone was able to bind to the nucleosome, consistent with a previous report[32] (Fig. 1c) and even in the absence of linker DNA (Supplementary Fig. 1b). The HMGA2-dependent mobility shift of the nucleosome complexes was also observed in the presence of H1.2, but their mobility was substantially slower as compared to the nucleosomes complexed with HMGA2 alone (Fig. 1c). Collectively, these results indicated that HMGA2 interacts with H1 indirectly via nucleosomes. The association of HMGA2 with H1 in the presence of nucleosomes appears to exclude the previously proposed possibility that HMGA proteins mediate chromatin decondensation by competing with H1 and excluding it from chromatin.

### HMGA2 promotes chromatin condensation in vitro

We next examined whether HMGA2 directly affects the extent of chromatin condensation in an in vitro reconstitution assay with the polynucleosome consisting of 12 nucleosomes assembled on the 12 tandem repeats of the Widom 601 DNA sequence (repeat length of 208 bp)[33]. An increase of the concentration of $MgCl_2$ resulted in aggregation (inter-polynucleosome association) of the polynucleosomes, which could be separated as sediments at the bottom of the reaction tube by centrifugation (Fig. 2a). We found that the addition of HMGA2 drastically reduced the $MgCl_2$ concentration required for aggregation of the polynucleosomes (Fig. 2a, b). The presence of both HMGA2 and H1.2 appeared to have an additive effect on polynucleosome aggregation (Fig. 2a, b). These results thus suggested that

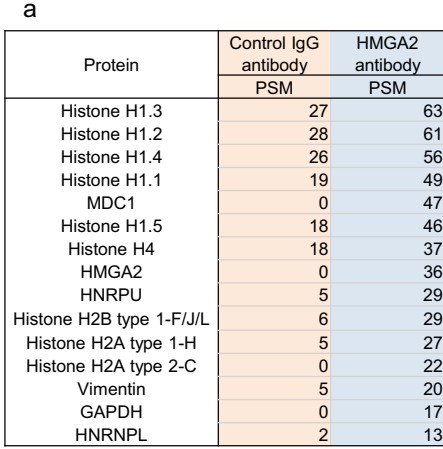

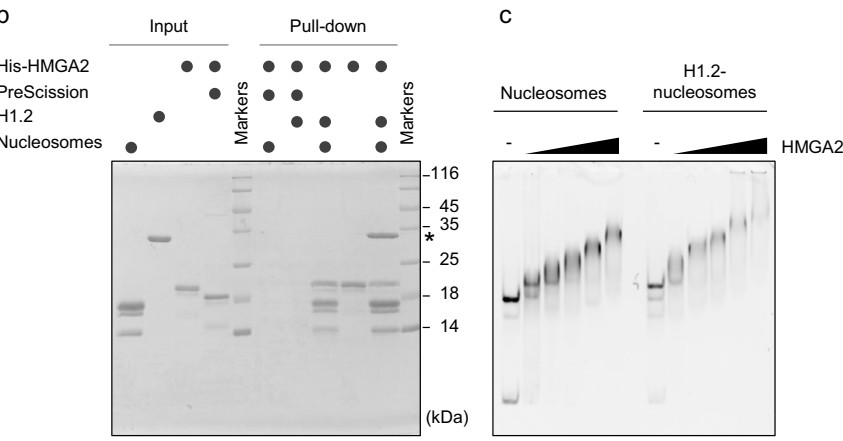

**Fig. 1 | HMGA2 forms a complex with histone H1. a** HMGA2-interacting proteins identified by IP-MS analysis. Endogenous HMGA2 was immunoprecipitated with antibodies to control immunoglobulin G (IgG) or HMGA2 (Cell Signaling, #8179) from a chromatin fraction of cells isolated from the mouse neocortex at E11.5. PSM, peptide spectrum match (Control: $n = 3$ independent experiments, HMGA2 antibody: $n = 2$ independent experiments). **b** SDS–polyacrylamide gel electrophoresis (PAGE) and Coomassie brilliant blue (CBB) staining of input samples (left) and pulldown samples (right) prepared with Ni-NTA beads after incubation of His-tagged human HMGA2 with recombinant human H1.2 and reconstituted mononucleosomes as indicated. His-HMGA2 treated with PreScission protease was examined as a control. Note that the nucleosomal core histones, but not the linker histone H1.2, were pulled down by His-tagged HMGA2 even in the absence of the nucleosome. The H1.2 band is indicated by an asterisk, and molecular mass markers are also shown ($n = 3$ independent experiments). **c** EMSA analysis of incubation mixtures containing increasing amounts of HMGA2 (0, 0.15, 0.25, 0.35, 0.45 µM) together with mononucleosomes (0.1 µM) in the absence or presence of H1.2 (0.7 µM). The native PAGE gel was stained with ethidium bromide to detect DNA ($n = 2$ independent experiments).

HMGA2 promotes condensation of chromatin in the absence or presence of H1.2.

We also investigated whether HMGA2 is able to promote intrapolynucleosome compaction through direct observation of polynucleosomes by atomic force microscopy (AFM). In the absence of HMGA2, polynucleosomes appeared as an open beaded chain of 12 nucleosomes connected by linker DNA (Fig. 2c). The addition of HMGA2 induced compaction of the polynucleosomes, with less space remaining between nucleosomes within each array (Fig. 2c). Indeed, the radius of the smallest circle that encloses a polynucleosome was $22.2 \pm 1.4\%$ ($P = 3.7 \times 10^{-16}$) reduced by the addition of HMGA2 (Fig. 2d), indicating that HMGA2 is able to directly promote chromatin condensation.

**HMGA2 protects linker DNA of nucleosomes in vitro**

We examined the effect of HMGA2 on chromatin accessibility to DNase I, which is dependent on the state of chromatin condensation. Incubation of the polynucleosomes consisting of 12 nucleosomes with DNase I yielded a DNA ladder as a result of the preferential accessibility and digestion of the linker DNA regions. The addition of H1.2, which binds linker DNAs and induces internucleosomal compaction, attenuated ladder formation (Fig. 2e). We found that the addition of HMGA2 protected the linker DNA digestion from DNase I (Fig. 2e). The linker DNA protection by HMGA2 was additively enhanced in the presence of H1.2 (Fig. 2e), suggesting that HMGA2 and H1 protect the linker DNAs in the polynucleosomes at different sites or through different mechanisms.

To test whether HMGA2 protects linker DNA even in the absence of internucleosomal interactions, we performed another chromatin accessibility assay with micrococcal nuclease (MNase), which preferentially cleaves linker DNA regions that are detached from histones, and with mononucleosomes that had been reconstituted with 193 bp of the Widom 601 DNA sequence and purified core histones (Fig. 2f). The mononucleosomes used contained 24 bp linker DNAs at both ends. Consistent with previous findings, the addition of H1.2, which protects linker DNA, resulted in the generation of longer DNA fragments after MNase treatment (Fig. 2f)[34]. We found that the addition of HMGA2 also resulted in the production of longer DNA fragments by MNase treatment, although fragment length differed from that

observed in the presence of H1.2 (Fig. 2f). These results suggest that HMGA2 and H1 protect different sites of linker DNA.

**HMGA2 localizes to heterochromatin in the mouse neocortex**

The effect of HMGA2 on chromatin condensation in vitro prompted us to investigate the genomic localization of endogenous HMGA2 in vivo. We therefore performed chromatin immunoprecipitation (ChIP)-sequencing (seq) analysis of endogenous HMGA2 in parallel with assay for transposase-accessible chromatin (ATAC)-seq and DNase I-seq analyses. This approach allowed us to compare HMGA2 deposition with chromatin accessibility in embryonic neocortical cells (Fig. 3a, b). The HMGA2 binding regions showed a higher AT content than did bulk DNA (HMGA2 binding regions; 71.9% ± 1.5%, Mean ± s.d.; bulk DNA; 58.9%), consistent with previous observations with other cell types[25,35]. Importantly, the HMGA2 binding regions showed a lower accessibility to DNase I, and Tn5, a lower level of H3K27me3 deposition (Fig. 3c, d and Supplementary Fig. 2a, b) and a higher level of HP1 deposition (Fig. 3e) compared with surrounding genomic regions (±2 kbp). HMGA2 binding regions tended to exclude promoters and were enriched with repeat sequences such as satellites (Supplementary Fig. 3a–c). We obtained essentially identical results with another preparation of antibodies to HMGA2 (Supplementary Fig. 3d–h). We also performed Hi-C analysis[36] of embryonic neocortical cells and found that HMGA2 binding regions were enriched in the B compartment (Fig. 3f and Supplementary Fig. 3i). These results together supported the notion that endogenous HMGA2 localizes to condensed chromatin or heterochromatin in mouse embryonic neocortical cells.

To visualize the global localization of HMGA2 in vivo, we established knock-in mice in which the coding sequence for enhanced green fluorescent protein (EGFP) was inserted into the endogenous *Hmga2* locus and detected EGFP fluorescence in the neocortex at E11.5 (Fig. 3g). Expression of the HMGA2-EGFP fusion protein was apparent at a high level mainly in the ventricular zone (Fig. 3g), where *Hmga2* is highly expressed. The HMGA2-EGFP signal appeared to overlap with Hoechst 33342, HP1, and H3K9me3 foci in cells located within the ventricular zone (Fig. 3g, Supplementary Fig. 3j). This overlap suggested that HMGA2 globally localizes to condensed chromatin in vivo.

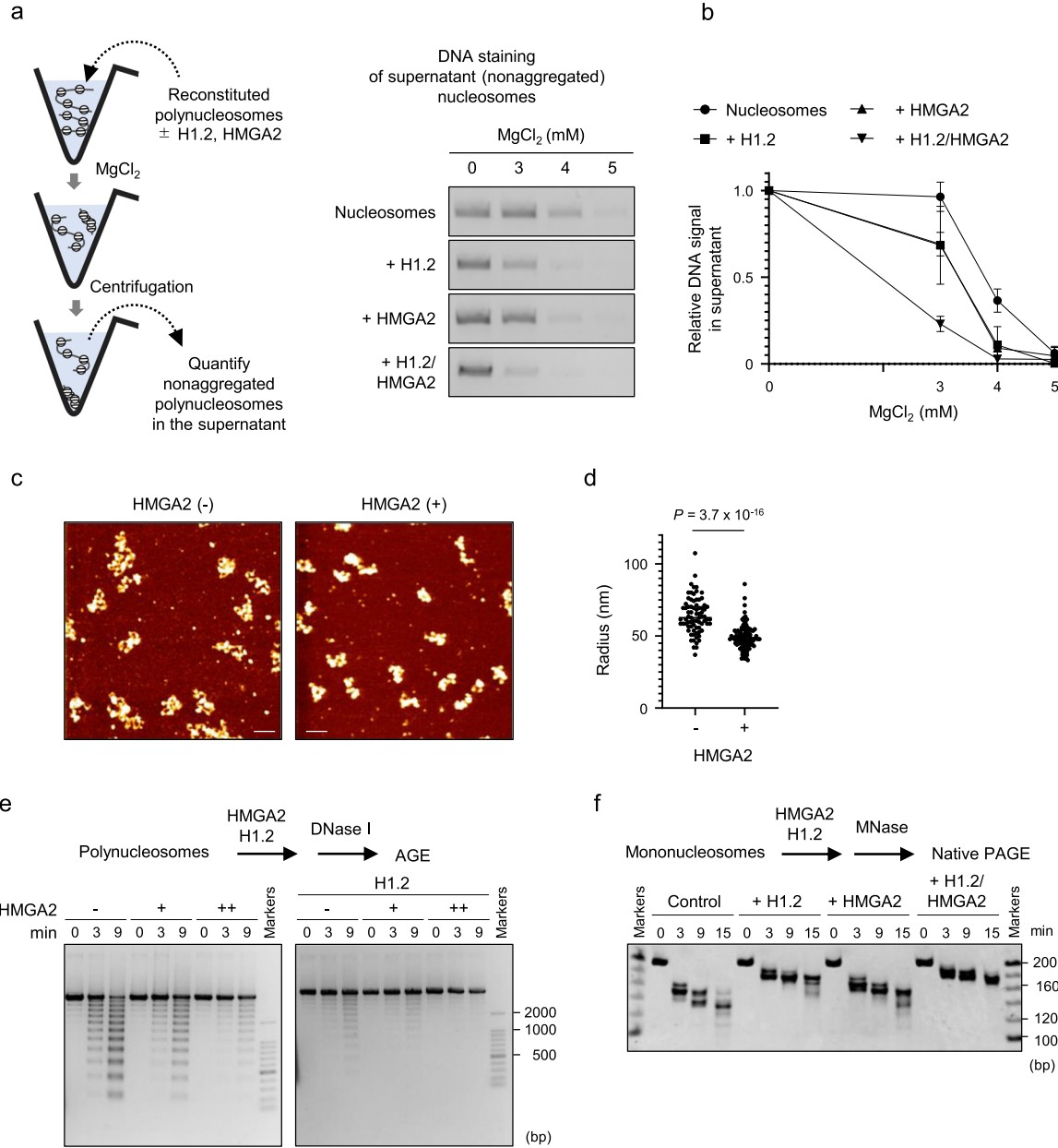

**Fig. 2 | HMGA2 promotes chromatin condensation in vitro. a** Scheme for the MgCl₂-mediated polynucleosome aggregation assay (left). Polynucleosomes reconstituted on the 12 tandem repeats of the Widom 601 DNA sequence were incubated with or without HMGA2 and H1.2 and in the presence of the indicated concentrations of MgCl₂ and were then isolated by centrifugation. The supernatant fraction containing nonaggregated polynucleosomes was subjected to agarose gel electrophoresis and staining of DNA with ethidium bromide (right). Note that the addition of H1.2 in this assay has been shown to promote aggregation of polynucleosome arrays, as indicated by a lowering of the concentration of MgCl₂ required for such aggregation[90]. **b** Quantification of nonaggregated polynucleosomes in experiments similar to that in (**a**). The DNA signal intensity was normalized by the maximum and minimum values. Data are means ± s.d. ($n = 3$ independent experiments). **c** Representative AFM topographic images of polynucleosomes reconstituted on the 12 tandem repeats of the Widom 601 DNA sequence and incubated with or without HMGA2. Scale bars, 100 nm.

**d** Quantification of the radius of the smallest circle encompassing individual polynucleosomes (control: $n = 76$, + HMGA2: $n = 105$) in AFM images as in (**c**). The mean values are indicated. Mann–Whitney U test (two-sided). **e** DNase I sensitivity assay for polynucleosomes. Polynucleosomes reconstituted on the 12 tandem repeats of the Widom 601 DNA sequence were incubated with or without HMGA2 in the absence (left) or presence (right) of H1.2 (5.0 μM) and then treated with DNase I for 0, 3 or 9 min. The reaction products were analyzed by agarose gel electrophoresis (AGE) and staining with ethidium bromide ($n = 3$ independent experiments). **f** MNase sensitivity assay for mononucleosomes. Mononucleosomes reconstituted with 193 bp of the Widom 601 DNA sequence and purified core histones were incubated without (Control) or with H1.2 alone, HMGA2 alone, or both H1.2 and HMGA2 and then treated with MNase for 0, 3, 9, or 15 min, after which the reaction products were analyzed by native PAGE followed by ethidium bromide staining ($n = 2$ independent experiments).

## HMGA2 forms droplets with nucleosomes

HMGA2 has a high intrinsic disorder score throughout its amino acid sequence, with such a high score often being associated with liquid-liquid phase separation (LLPS) (Fig. 4a). Given that chromatin condensation and gene repression mediated by HP1, MeCP2, and PcG proteins have been proposed to involve LLPS[37–43], we examined whether HMGA2 also undergoes LLPS. We first asked whether the foci of HMGA2-EGFP in neocortical cells are dynamic. We indeed found that these foci showed a rapid recovery in a FRAP assay (Fig. 4b, c), suggesting that they are dynamic and possibly reflect LLPS.

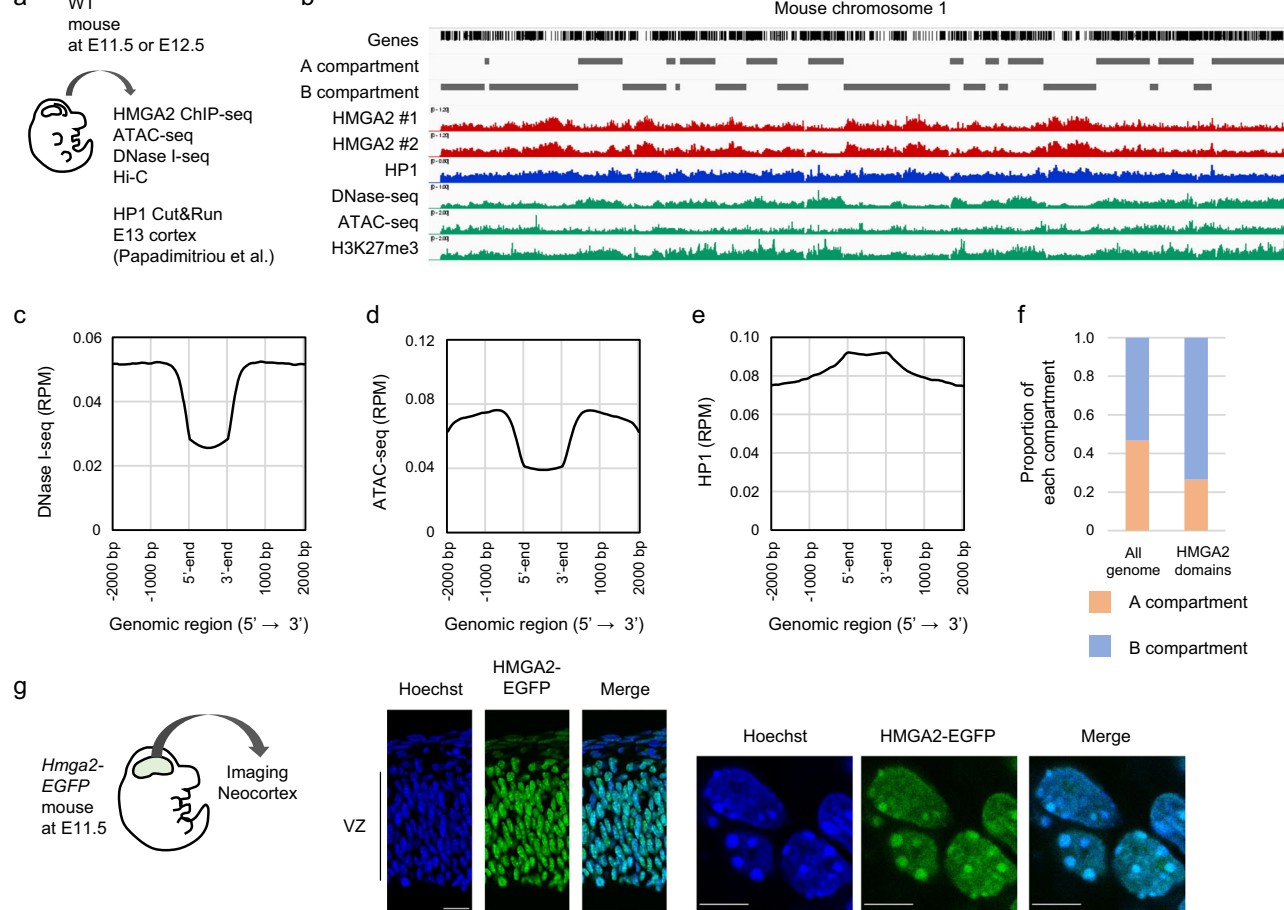

**Fig. 3 | HMGA2 preferentially localizes to heterochromatin in vivo. a** Neocortical cells isolated from wild-type (WT) mouse embryos were subjected to ChIP-seq analysis with antibodies to HMGA2 (E11.5), ATAC-seq analysis (E11.5), DNase I-seq analysis (E12.5), Cut&Run analysis (E13.5, data obtained from Papadimitriou et al.[91]) and Hi-C analysis (E11.5). HMGA2-bound genomic regions (HMGA2 domains) were defined by peak calling with the use of F-seq software[72]. **b** A representative track showing the distribution of HMGA2, HP1, ATAC-seq, DNase-seq and H3K27me3 signals and A/B compartment across the mouse chromosome 1. HMGA2 #1 (Cell Signaling, #8179), HMGA2 #2 (in-house antibodies). **c** Representative plot of DNase I-seq signals (RPM, read count per million mapped reads) around HMGA2 domains. HMGA2 domains (5′ to 3′) are indicated at the center of the x-axis (n = 2

independent experiments). **d** Representative plot of ATAC-seq signals (RPM) around HMGA2 domains. HMGA2 domains (5′ to 3′) are indicated at the center of the x-axis (n = 4 independent experiments). **e** Representative plot of HP1 signals (RPM) around HMGA2 domains. HMGA2 domains (5′ to 3′) are indicated at the center of the x-axis. **f** Compartment (principal component 1) distribution for HMGA2 domains in the Hi-C data. A and B compartments were defined by principal component analysis with HiCExplorer. **g** Fluorescence microscopy of a coronal section of the *Hmga2-EGFP* mouse neocortex at E11.5 (left, scale bar = 20 μm) as well as of a portion of the ventricular zone (VZ) also showing Hoechst 33342 staining (right, scale bars = 5 μm) (n = 3 independent experiments).

We then investigated whether purified HMGA2 forms droplets in solution. An in vitro assay indeed revealed that recombinant HMGA2 formed spherical droplets in the presence of mononucleosomes, whereas HMGA2 alone or mononucleosomes alone did not form droplets under the same condition (Fig. 4d, e). The droplet forming ability depended on HMGA2 and salt concentrations, which is a typical feature of LLPS (Fig. 4f). The concentration of HMGA2 protein necessary for droplet formation was comparable to that of other proteins reported to undergo LLPS[37,40]. HMGA2 labeled with the ATTO647 fluorescent tag was also found to be incorporated into the droplets formed by HMGA2 and mononucleosomes and to undergo rapid recovery in a FRAP assay, indictive of dynamic movement and exchange of HMGA2 between the droplets and the medium (Fig. 4g, h). These results thus suggested that HMGA2 undergoes LLPS together with nucleosomes.

### Localization of HMGA2 to the gene body is associated with gene repression

Chromatin condensation is generally associated with gene repression. We therefore investigated how HMGA proteins might regulate gene expression at HMGA-bound regions in isolated NPCs. To this end, we

deleted *Hmga2* or both *Hmga1* and *Hmga2* with the use of the *Sox1-Cre* transgene[44,45] and performed RNA-seq analysis of NPCs isolated from the E12.5 neocortex of control or conditional knockout (cKO) mice by fluorescence-activated cell sorting (FACS) as CD133high cells (Fig. 5a). Examination of the level of HMGA2 binding to the loci of DEGs, including 3 kbp of both 5′ and 3′ flanking regions of each gene (for this analysis, a gene begins at the transcription start site [TSS] and ends at the transcription termination site [TTS]), revealed that DEGs whose expression was upregulated in *Hmga1/2* cKO cells showed more HMGA2 binding to the gene body region as well as to upstream and downstream regions compared with all genes or downregulated DEGs (Fig. 5b, c and Supplementary Fig. 4a, b). Similarly, DEGs whose expression was upregulated in *Hmga2* cKO cells showed a higher level of HMGA2 binding to the gene body region than did all genes or the corresponding downregulated DEGs (Fig. 5d, e and Supplementary Fig. 4c, d). These results suggested that localization of HMGA2 to the gene body is associated with gene repression.

We then categorized HMGA2-bound genes according to its binding patterns. Analysis by k-means clustering of all genes resulted in five clusters characterized by high levels of HMGA2 binding at the

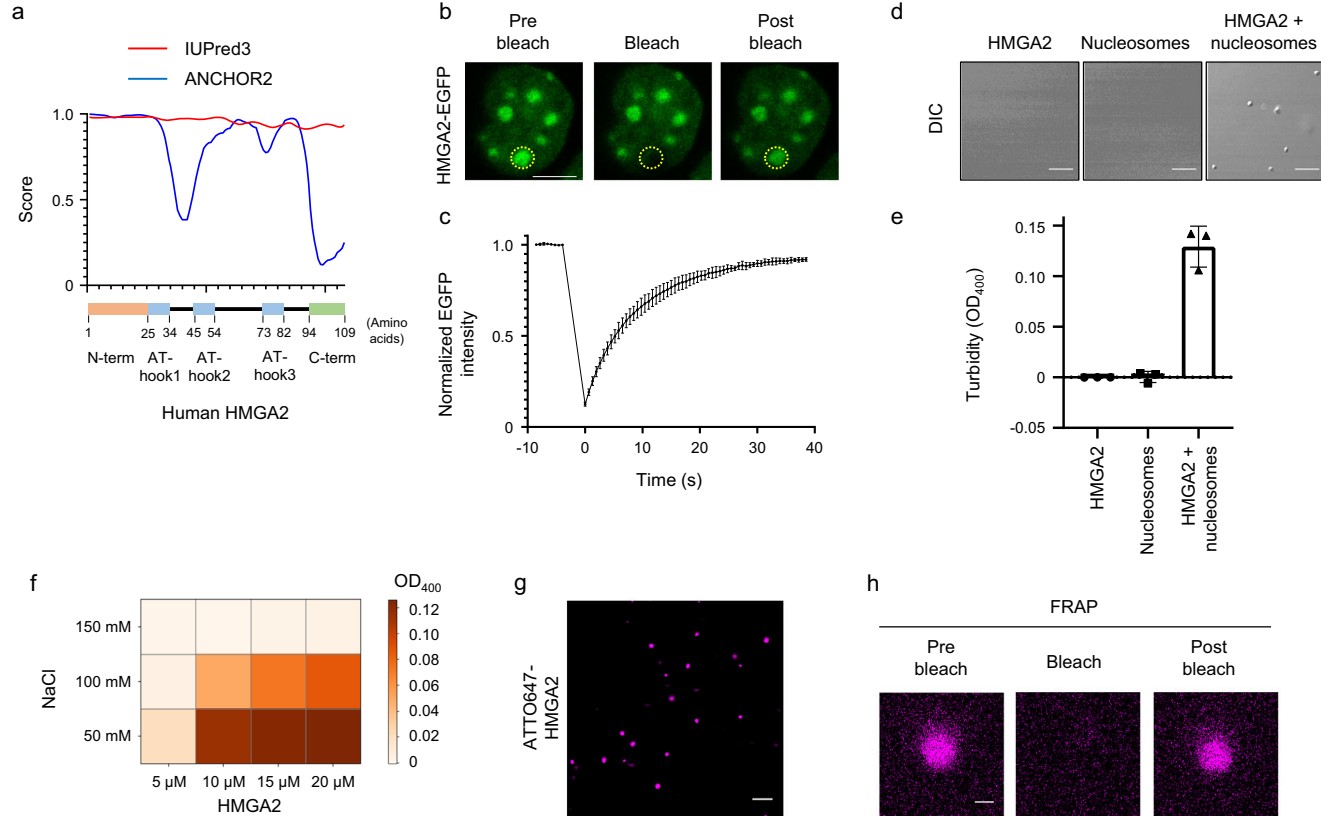

**Fig. 4 | HMGA2 forms droplets with nucleosomes. a** The disorder score was calculated with IUPred3 or ANCHOR2[92,93]. **b** FRAP analysis of HMGA2-EGFP in a neocortical cell prepared from the *Hmga2-EGFP* mouse. The EGFP signal at the bleached focus (indicated by the dotted circle) was mostly recovered by 40 s after bleaching (postbleach). Scale bar, 5 μm. **c** Quantification of EGFP signal intensity at the bleached focus in FRAP experiments as in (**b**) (photobleaching ended at $t = 0$). Data are means ± s.d. ($n = 72$ droplets from 72 cells, $n = 3$ independent experiments). **d** Droplet formation analysis. Recombinant HMGA (20 μM) and mononucleosomes (800 nM) were incubated alone or together in a droplet formation solution and then observed with a differential interference contrast (DIC) microscope. Scale bars, 10 μm. **e** Turbidity of incubation mixtures as in (**d**) was assessed

by measurement of optical density at 400 nm (OD$_{400}$). Data are means ± s.d. ($n = 3$ independent experiments). **f** Phase diagram of HMGA2 droplet formation. HGMA2 at the indicated concentration was added to a droplet formation buffer containing the indicated concentration of NaCl and OD$_{400}$ was measured after 10 min ($n = 3$ independent experiments). **g**, **h** Droplet formation by and FRAP analysis of HMGA2 labeled with a fluorescent tag. Fluorescence microscopy of droplets formed by 20 μM ATTO647-labeled recombinant HMGA2 (labeling efficiency of <30%) and 800 nM mononucleosomes is shown in (**g**). Scale bar, 10 μm. Such droplets were subjected to FRAP analysis (postbleach corresponds to 55 s after bleaching), with the signal intensity having recovered to 80.2% ± 2.0% (mean ± s.d., $n = 9$ from three independent experiments) by 30 s after bleaching (**h**). Scale bar, 1 μm.

promoter (clusters 1 and 3), the gene body (cluster 2), or the TTS (cluster 4) regions or by an overall low level of HMGA2 binding (cluster 5) (Fig. 5f; see also Supplementary Fig. 4e for results obtained with different HMGA2 antibodies). The expression levels of cluster 2 genes were significantly upregulated in *Hmga1/2* cKO cells (Fig. 5g; see also Supplementary Fig. 4f) and in *Hmga2* cKO cells (Supplementary Fig. 5). Cluster 2 genes showed a higher level of expression compared with genes of other clusters in control NPCs and an enrichment of neuronal genes (Fig. 5h, i). Together, these results indicated that binding of HMGA2 to the gene body of highly expressed and neuronal genes is associated with gene repression in embryonic neocortical NPCs.

**The AT-hook1 domain is necessary for chromatin condensation by HMGA2 in vitro and in cellulo**

To distinguish between chromatin condensation-dependent and -independent roles of HMGA2, we attempted to generate an HMGA2 mutant that retains the ability to bind to nucleosomes but lacks that to promote chromatin condensation. We purified a series of recombinant HMGA2 mutant proteins lacking the NH$_2$-terminal, AT-hook1, AT-hook2, AT-hook3 or COOH-terminal (N del, hook1 del, hook2 del, hook3 del and C del, respectively) and performed an aggregation assay with polynucleosomes (Fig. 6a). The N del and C del mutants reduced the fraction of unaggregated polynucleosomes in the presence of

4 mM MgCl$_2$, indicating that the NH$_2$-terminal and COOH-terminal domains are dispensable for HMGA2-induced chromatin condensation (Fig. 6b, c). In contrast, deletion of either hook1 or 3 completely and that of hook2 partially reduced the chromatin aggregation activity of HMGA2. These suggested that the all AT-hook domains, especially hook1 and hook3, are important for the ability of HMGA2 to promote chromatin condensation. Consistent with this observation, AFM analysis also showed that the N del mutant reduced the radius of polynucleosomes, while the hook1, hook2 and hook3 mutants did not significantly reduce it (Fig. 6d, e). Given that AT-hook domains have been shown to bind to DNA[46], we examined whether the AT-hook del mutants of HMGA2 are able to bind to nucleosomes. We incubated the reconstituted mononucleosome with the full-length or mutant forms of HMGA2 and then subjected the incubation mixtures to EMSA analysis, and found that all the single AT-hook del mutants were capable of inducing a nucleosome band shift as effectively as the full-length HMGA2 (Fig. 6f). We also performed ChIP analysis in vivo and found that the hook1 del mutant of HMGA2 exhibits comparable binding to nucleosomes as the full-length HMGA2 (Supplementary Fig. 6a). Each of the AT-hook domains thus appears to be necessary for induction of chromatin condensation but dispensable for nucleosome binding,

We then asked how HMGA2 promotes chromatin condensation in an AT-hook domain-dependent manner. We hypothesized that

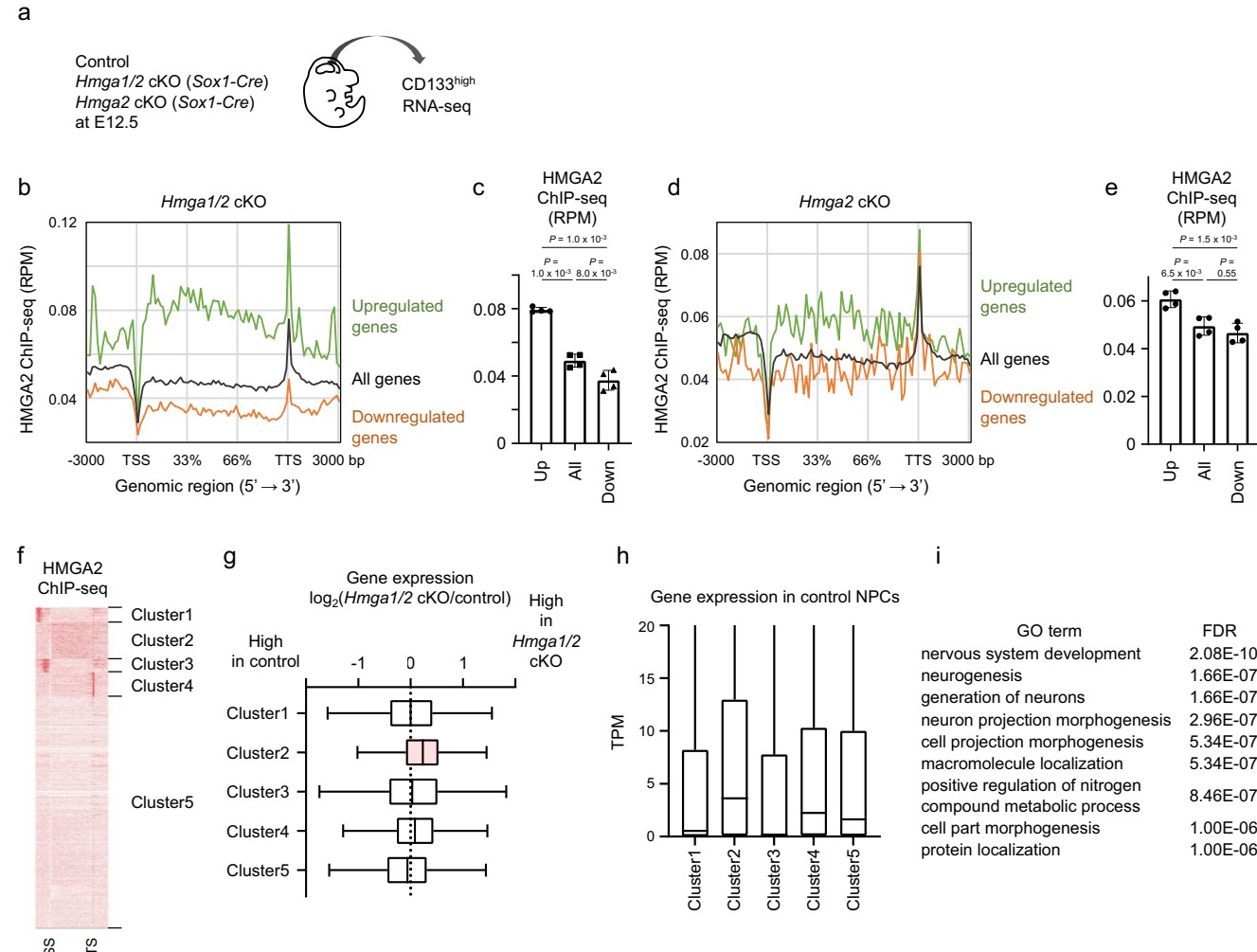

**Fig. 5 | Gene body localization of HMGA2 is associated with gene repression.**
**a** RNA isolated from CD133[high] NPCs derived from the neocortex of control, *Hmga2* cKO, or *Hmga1/2* cKO mice at E12.5 was subjected to RNA-seq analysis.
**b** Representative plot of averaged HMGA2 ChIP-seq signals around the gene body of all genes (black), genes upregulated by *Hmga1/2* cKO (green, 1266 genes), and genes downregulated by *Hmga1/2* cKO (orange, 1826 genes). TSS, transcription start site; TTS, transcription termination site. **c** Average signals at the gene body were calculated for each gene set in (**b**). Data are means ± s.d. (*n* = 4 independent experiments). Tukey's multiple comparison test. **d** Representative plot of averaged HMGA2 ChIP-seq signals around the gene body of all genes (black), genes upregulated by *Hmga2* cKO (green, 518 genes), and genes downregulated by *Hmga2* cKO (orange, 341 genes). **e** Average signals at the gene body were calculated for

each gene set in (**d**). Data are mean ± s.d. (*n* = 4 independent experiments). Tukey's multiple comparison test. **f** HMGA2 binding patterns were clustered by ngsplot (*k*-means clustering, *k* = 5). The HMGA2 ChIP-seq signals ranging from 3000 bp upstream of the TSS to 3000 bp downstream of the TTS are shown as a heat map. **g** Differences in gene expression for each cluster between neocortical NPCs of *Hmga1/2* cKO mice and those of control mice. Data are presented as box plots, with the boxes representing the median and upper and lower quartiles and the whiskers indicating the range (*n* = 3 independent experiments). **h** Gene expression levels (TPM, transcripts per million) for each cluster in neocortical NPCs of control mice. Data are presented as box plots, with the boxes representing the median and upper and lower quartiles and the whiskers indicating the range (*n* = 3 independent experiments). **i** Gene ontology analysis of upregulated in *Hmga1/2* cKO mice.

HMGA2 might affect the flexibility of linker DNA, which is known to affect chromatin condensation. To investigate the flexibility of linker DNA, we reconstituted nucleosomes with fluorescent labels (Cy3 or Cy5) attached to each end of the linker DNA in order to perform a fluorescence resonance energy transfer (FRET) assay. The addition of full-length HMGA2, but not the hook del mutants, increased the FRET signal (Fig. 6g, h). We confirmed that both full-length and hook del mutants of HMGA2 showed similar binding affinities for the fluorescently labeled nucleosomes by EMSA analysis (Supplementary Fig. 6b). These results suggested that HMGA2 restricts the movement of linker DNA in an AT-hook- dependent manner, which may underlie its ability to induce chromatin condensation.

In addition to these in vitro assays, we also investigated whether AT-hook1 is important for HMGA2-induced chromatin condensation in cellulo by forcibly expressing full-length or hook1 del

forms of HMGA2 tagged with GFP in IMR90 cells (fibroblasts isolated from normal human lung tissue). Observation of nuclear structure revealed that expression of HMGA2-GFP induced the formation of aggregates containing HMGA2-GFP and DNA (Fig. 6i, j), consistent with previous findings[26,27]. In contrast, expression of the GFP-tagged hook1 del mutant did not induce the formation of such aggregates (Fig. 6i, j), suggesting that the AT-hook1 domain is important for the ability of HMGA2 to induce chromatin condensation in cellulo.

We then tested whether the AT-hook1 domain is required for droplet formation by HMGA2 and nucleosomes. We indeed found that, in contrast to the full-length protein, the hook1 del mutant of HMGA2 did not form droplets even in the presence of the mononucleosome (Fig. 6k, l). By contrast, the C del mutant of HMGA2 formed droplets even more efficiently than did the full-length protein (Fig. 6k, l). Given

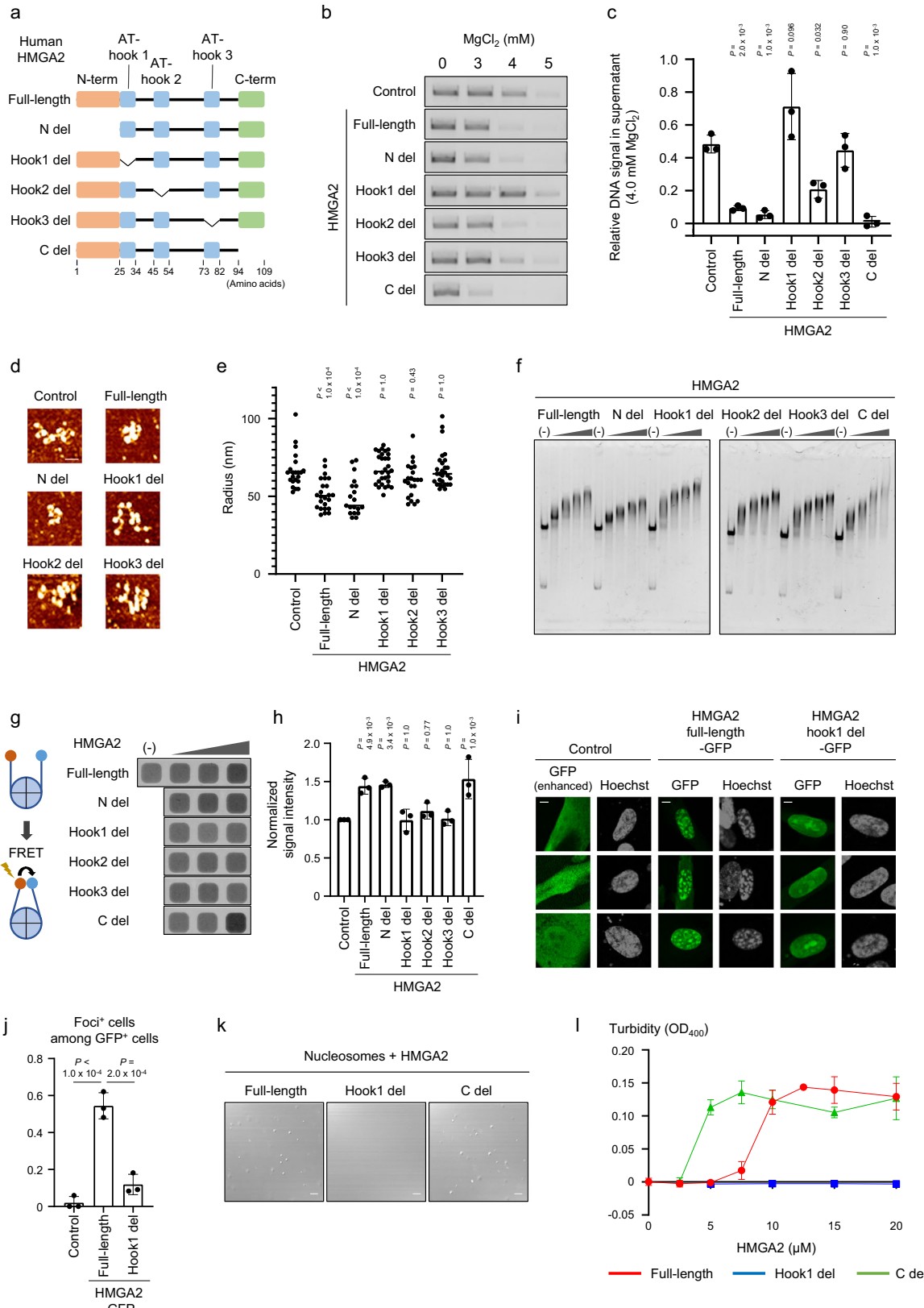

the proposed intramolecular interaction between the COOH-terminal domain and the AT-hook domains of HMGA2[47], it is possible that the COOH-terminal domain masks the AT-hook domains and thereby impedes LLPS. These results thus suggested that the AT-hook1 domain, but not the COOH-terminal domain, is particularly important for droplet formation by HMGA2 and nucleosomes.

## HMGA2 maintains neurogenic progenitors via condensate formation

HMGA2 has been shown to promote NPC proliferation and neuronal fate commitment in the developing mouse neocortex[6,8]. We therefore asked whether AT-hook1-dependent chromatin condensation contributes to these functions of HMGA2. We introduced a plasmid

**Fig. 6 | The AT-hook 1 domain of HMGA2 is necessary for HMGA2-induced chromatin condensation in vitro and in cellulo. a** Domain organization and constructed deletion mutants of HMGA2. **b**, **c** $MgCl_2$-dependent polynucleosome aggregation assay performed as in Fig. 2a with HMGA2 deletion mutants. Representative results (**b**) and quantification of data from three independent experiments (**c**) are shown. Quantitative data are means ± s.d. and were normalized by the maximum and minimum values. Dunnett's multiple comparison test.
**d** Representative AFM topographic images of polynucleosomes reconstituted on the 12 tandem repeats of the Widom 601 DNA sequence and incubated with the mutant forms of HMGA2. Scale bars, 50 nm. **e** Quantification of the radius of the smallest circle encompassing individual polynucleosomes (control (without HMGA2): $n = 22$, + full-length HMGA2: $n = 24$, N del: $n = 19$, hook1 del: $n = 30$, hook2 del: $n = 22$, hook3 del: $n = 29$) in AFM images as in (**c**). The mean values are indicated. Mann–Whitney U test (two-sided). **f** EMSA analysis of the binding of HMGA2

deletion mutants to mononucleosomes performed as in Fig. 1c. **g** FRET assay of linker flexibility. **h** Quantification of relative FRET signal intensity for experiments similar to that in (**d**). Data are means ± s.d. ($n = 3$ independent experiments). Dunnett's multiple comparison test. **i** Fluorescence microscopic analysis of focus formation by GFP, GFP-tagged full-length or hook1 del forms of HMGA2 expressed in IMR90 cells. DNA was stained with Hoechst 33342. Scale bars, 5 μm.
**j** Quantification of the proportion of GFP⁺ cells with foci in images similar to those in (**h**). Data are means ± s.d. ($n = 3$ independent experiments). Two-tailed Student's $t$ test. **k** Analysis of droplet formation by full-length, hook1 del, or C del forms of recombinant HMGA2 (20 μM) and mononucleosomes (800 nM) as in Fig. 4d. Scale bars, 10 μm. **l** Quantification of turbidity in experiments performed as in (**k**) with various concentrations of the HMGA2 proteins. Data are means ± s.d. ($n = 3$ independent experiments).

encoding GFP alone (control) or together with a plasmid encoding full-length or hook1 del mutant forms of HMGA2 into NPCs of the mouse embryonic neocortex by electroporation at E15.5 (Fig. 7a). Immunohistofluorescence analysis of the brain at postnatal day (P) 1 revealed that most GFP-positive cells in control samples had undergone neuronal differentiation and migration into the cortical plate (CP) (Fig. 7a, d). In contrast, overexpression of full-length HMGA2 reduced the fraction of GFP-positive cells in the CP and increased that of those in the intermediate zone (IMZ) and the ventricular zone (VZ)/subventricular zone (SVZ) (Fig. 7d, e). Moreover, overexpression of full-length HMGA2 resulted in a significant increase in the proportion of GFP-positive cells that were also positive for the proliferation marker Ki67 (Fig. 7d, e and Supplementary Fig. 7). Expression of the hook1 del mutant of HMGA2 did not affect the proportion of these cells positive for Ki67 (Fig. 7d, e and Supplementary Fig. 7), indicating that HMGA2 promotes progenitor maintenance in a manner dependent on the AT-hook1 domain. We then asked whether the AT-hook1-dependent functions of HMGA2 are ascribable to condensate formation via LLPS. We thus examined if the intrinsically disordered domain of PUB1 (PUB1₍IDR₎), which exhibits the condensate-forming activity[48], can rescue the loss of HMGA2's functions in the absence of the AT-hook1 domain. We found that a HMGA2 hook1 del mutant fused with PUB1₍IDR₎ at the N-terminus (PUB1₍IDR₎-HMGA2 hook1 del) formed droplets in vitro (Fig. 7b, c). Importantly, overexpression of PUB1₍IDR₎-HMGA2 hook1 del mutant increased the proportion of cells in the IMZ and the VZ/SVZ, and decreased the proportion of cells in the CP, which phenocopied full-length HMGA2 (Fig. 7d, e). These results suggested that HMGA2 promotes progenitor maintenance via condensate formation.

We then examined which progenitor types (proliferative populations) are affected by HMGA2 overexpression. Expression of full-length HMGA2 from E15.5 increased the proportion of Tbr2-positive cells (generally considered as intermediate neuronal progenitors, or INPs) and that of NeuroD1-positive cells (immature neurons) among all GFP-positive cells at P1, whereas it did not significantly affect that of Sox2-positive cells (NPCs) (Fig. 7f). Again, these effects of full-length HMGA2 were not reproduced by the hook1 del mutant (Fig. 7f). Together, these results suggested that HMGA2 promotes the maintenance of neurogenic progenitors—in particular, INPs—in an AT-hook1 domain-dependent manner.

We also performed RNA-seq analysis of the embryonic neocortex subjected to electroporation with the plasmids for full-length or hook1 del mutant forms of HMGA2 at E15.5. Relatively undifferentiated cells including NPCs and INPs were isolated as CD133^high CD24^low cells positive for GFP from the neocortex at P1 by FACS, and subjected to RNA-seq analysis. We found that overexpression of full-length HMGA2 increased the levels of *Eomes* (the gene encoding Tbr2), *Neurod1*, and *Neurod6* mRNAs, and that these effects were again largely suppressed by deletion of the AT-hook1 domain, reflecting the increase of INPs (Fig. 7g). Furthermore, gene ontology (GO) analysis revealed that genes whose expression was upregulated by overexpression of full-

length HMGA2 were enriched in those related to the cell cycle including *Mki67* (the gene for Ki67), *Ccnd1*, *Ccnd2*, *Ccnd3*, *Cdk1*, *Cdk2*, and *Cdk4* (Fig. 7g, h). The expression of these genes was not significantly affected by expression of the hook1 del mutant of HMGA2. These results are consistent with those of the immunohistofluorescence analysis described above and support the notion that HMGA2 promotes the maintenance of proliferating cells including neurogenic progenitors. Of note, the expression levels of *Igf2bp2* and *Plag1*, both of which are HMGA target genes related to neurogenic fate commitment of neocortical progenitors[15,16] and were confirmed to be repressed by cKO of *Hmga1* and *Hmga2* in neocortical NPCs (Supplementary Fig. 8a) and significantly upregulated by overexpression of full-length HMGA2 but not by expression of the hook1 del mutant (Supplementary Fig. 8b).

We then asked whether HMGA2 also affects gliogenic fate in a manner dependent on the AT-hook1 domain. Our RNA-seq analysis revealed that overexpression of full-length HMGA2 in the neocortex from E15.5 attenuated expression of the astrocyte marker genes *Gfap*, *Aldh1l1*, *Sox9*, and *Fabp7* apparent at P1, whereas the hook1 del mutant had no such effect (Fig. 7g). These results suggested that HMGA2 promotes neurogenic fate at the expense of astrocytic fate in an AT-hook1-dependent manner.

Finally, we examined global changes in gene expression induced by HMGA2. Our RNA-seq data showed that the overall (averaged) expression levels of genes upregulated or downregulated as a result of full-length HMGA2 overexpression were not affected by expression of the hook1 del mutant (Fig. 7i), suggesting that the AT-hook1 domain is important for HMGA2-dependent changes in gene expression at the genome-wide level in neocortical NPCs. Importantly, PUB1₍IDR₎ fusion partially rescued the transcriptional changes caused by the hook1 deletion (Fig. 7i). Together, our results thus indicated that the hook 1 domain, which plays a key role in chromatin condensation by HMGA2, is necessary for HMGA2-mediated maintenance of neuronal progenitors in the developing neocortex.

## Discussion
By performing in vitro reconstitution analyses, we here unveiled that HMGA2 has an intrinsic ability to mediate chromatin condensation. Recombinant HMGA2 thus promoted aggregation of polynucleosomes in the presence of $MgCl_2$ as revealed by a sedimentation assay, induced intraarray compaction of polynucleosomes as revealed by AFM, and rendered polynucleosomes and mononucleosomes inaccessible to nucleases as revealed by DNase I and MNase sensitivity assays, respectively. Consistent with the presence of intrinsically disordered regions (IDRs) in its primary structure, HMGA2 was also able to form droplets (condensates) together with mononucleosomes. Furthermore, our examination of the genomic distribution of endogenous HMGA2 in vivo (in neocortical cells) revealed that HMGA2-bound genomic regions are inaccessible to DNase I and to Tn5 and are enriched in regions of the B compartment, consistent with the

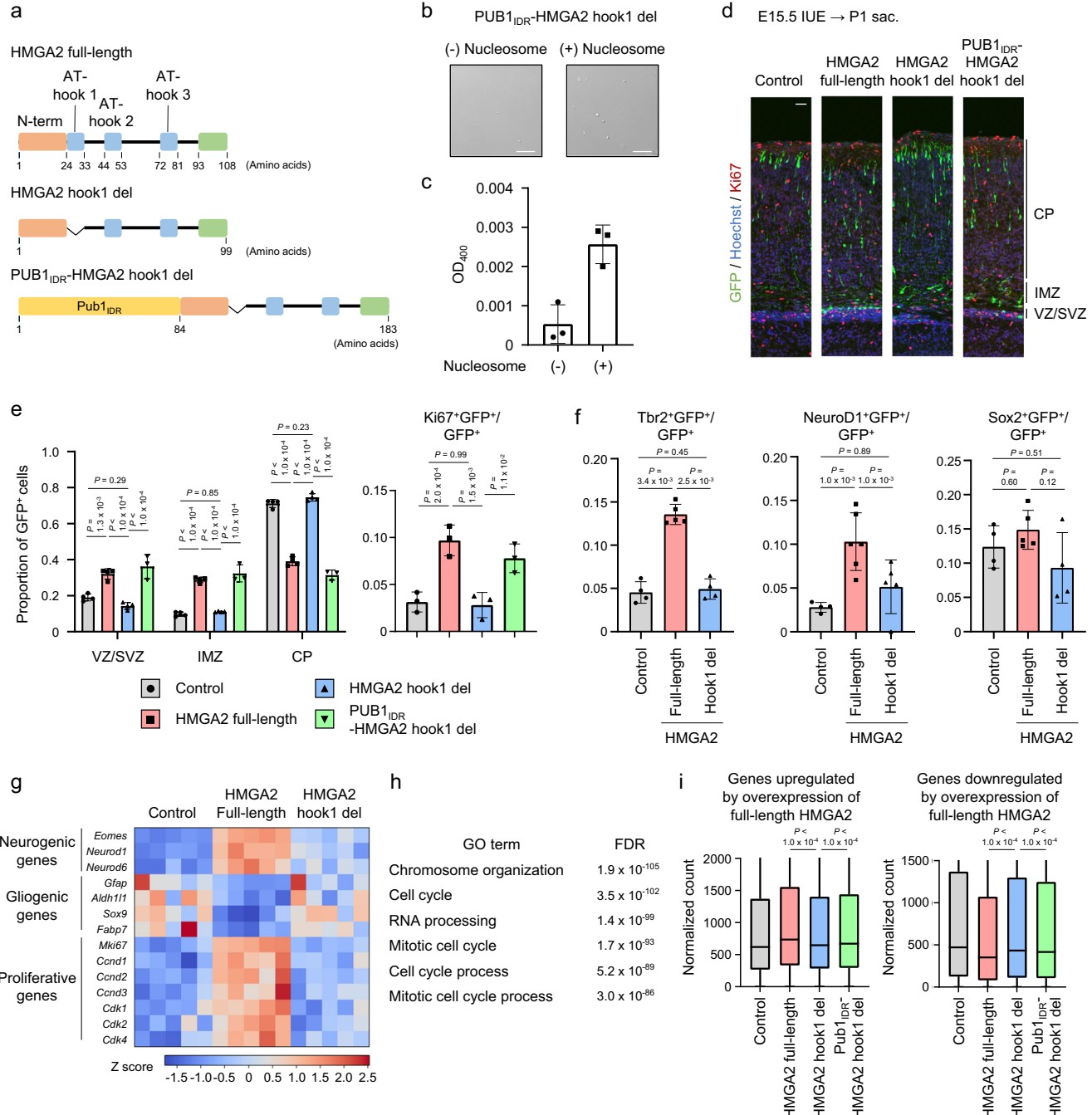

**Fig. 7 | HMGA2 regulates neuronal cell fate in an AT-hook 1-dependent manner.** **a** Domain organization and constructed mutants of HMGA2. **b** Analysis of droplet formation by recombinant PUB1_{IDR}-HMGA2 hook1 del (7.2 μM) in the presence or absence of mononucleosomes (800 nM) in a droplet formation solution and then observed with a DIC microscope. Scale bars, 10 μm. **c** Turbidity of incubation mixtures as in (**b**) was assessed by measurement of optical density at 400 nm (OD_{400}). Data are means ± s.d. (n = 3 independent experiments). **d** Immunohistofluorescence analysis of GFP and Ki67 in the neocortex of mice sacrificed (sac) at P1 after in utero electroporation (IUE) at E15.5 with an expression plasmid indicated in figure. Scale bar, 30 μm. **e** (left) Proportion of GFP-positive cells located in the cortical plate (CP), the intermediate zone (IMZ), or the ventricular zone (VZ)/subventricular zone (SVZ) in images as in (**d**). (right) Proportion of Ki67+ cells among GFP+ cells in images as in (**d**). Data are means ± s.d. (Localization: Control, HMGA2 full-length and HMGA2 hook1 del: n = 4, PUB1_{IDR}-HMGA2

hook1 del: n = 3, Ki67 staining: n = 3). Data are means ± s.d. Tukey's multiple comparison test. **f** Proportion of Tbr2+, NeuroD1+, or Sox2+ cells among GFP+ cells detected by immunohistofluorescence analysis under the same conditions as in (**d**). Data are means ± s.d. (Control, HMGA2 hook1 del: n = 4, HMGA2 full-length: n = 4 (Tbr2), 6 (NeuroD1), 5 (Sox2)). Tukey's multiple comparison test. **g** Expression of the indicated genes determined by RNA-seq from the neocortex at P1 after IUE at E15.5. (n = 5 independent experiments). **h** Enriched GO terms and their false discovery rate (FDR) values determined by functional annotation of genes whose expression was upregulated by overexpression of full-length HMGA2 in neocortical NPCs as measured by RNA-seq analysis. **i** Expression levels of all genes upregulated (left) or downregulated (right) by overexpression of full-length HMGA2 as determined by RNA-seq analysis. Data are presented as box plots, with the boxes representing the median and upper and lower quartiles and the whiskers indicating the range. Tukey's multiple comparison test (two-sided).

chromatin condensation activity of HMGA2. Importantly, we found that each of the AT-hook domains is necessary for this activity but dispensable for the DNA binding activity of HMGA2, and that deletion of AT-hook1 impairs the function of HMGA2 in regulation of gene expression patterns and maintenance of neurogenic progenitors in the developing neocortex. These results indicate that HMGA2 is a bona fide chromatin condensation factor and that it may regulate neural progenitor cell fate in a manner dependent on its chromatin condensation activity.

HMGA2 is unique among previously-identified nonhistone chromatin condensation factors. HP1, MeCP2, and PcG proteins bind to polynucleosomes harboring specific epigenetic modifications (H3K9me2/me3 for HP1, methylated DNA for MeCP2, and H3K27me3 and H2AK119ub for PcG proteins)[49–51], whereas we found that, like the linker histone H1, HMGA2 is able to bind to "bare" polynucleosomes without such modifications and in the absence of additional factors. Moreover, we found that HMGA2 and H1.2 probably bind to different positions of nucleosomes, as revealed by differential protection of linker DNA from MNase by these proteins. Consistent with this finding, HMGA2 and H1.2 were able to bind to nucleosomes simultaneously and to act cooperatively to condense chromatin. Given that linker length has been reported to control the binding affinity of transcription factors and chromatin modifying factors[52,53], H1 and HMGA2 may differentially regulate deposition of these factors to chromatin.

Several mechanisms, which are not mutually exclusive, may account for HMGA2-mediated chromatin condensation. We found that the flexibility of linker DNA was reduced by HMGA2 in an AT-hook1 domain-dependent manner by FRET analysis of mononucleosomes. Given that restriction of linker DNA flexibility has been proposed to mediate chromatin condensation by H1[54], this activity of HMGA2 may also account for its ability to mediate chromatin condensation. HMGA2 appears to be able to bind to multiple DNA sites via its three AT-hook domains, with this property possibly underlying its restriction of linker DNA movement.

Another possible mechanism for chromatin condensation mediated by HMGA2 is LLPS. We found that recombinant HMGA2 forms droplets in the presence of the mononucleosome and that this ability is dependent on the AT-hook1 domain. These results are significantly different from a recent report showing that HMGA1 protein alone can form droplets[55]. Furthermore, they showed in their paper that the C-terminal domain is essential for droplet formation[55], but our results showed that this domain is dispensable. Rather, our results showed that the AT-hook domains (especially hook1) are essential for droplet formation. Given that deletion of the AT-hook 1 domain reduced the functions of HMGA in regulation of NPC fate (Fig. 7), the AT-hook1 domain-dependent LLPS found in this study appears to be more physiologically relevant in this context.

HMGA2 may promote LLPS as a result not only of its IDRs, which constitute most of the protein, but also of an ability to remodel the nucleosome core structure. A recent study found that HP1 reshapes the structure of the nucleosome core and thereby exposes buried nucleosomal regions and that this reshaping contributes to LLPS of chromatin[56]. Of note, we found that HMGA2 reduced the stability of H2A-H2B dimers and H3-H4 tetramers in a thermal stability assay (Supplementary Fig. 9), suggesting that HMGA2 might promote LLPS also by reshaping the nucleosome core.

Unexpectedly, cKO of both Hmga1 and Hmga2 in NPCs tended to increase the expression of highly expressed genes that harbor HMGA2 at the gene body, but not that of genes with HMGA2 bound at the promoter, indicative of a gene body-specific repressive function of HMGA proteins. It will be of interest to investigate the mechanism responsible for recruitment of HMGA proteins to the gene body of specific genes as well as the role of such binding in future studies.

HMGA proteins have been shown not only to repress gene expression but also to activate it in a context-dependent manner[9–17].

We found that HMGA2 increased the expression of Igf2bp2 and Plag1, both of which contribute to the neurogenic fate of NPCs, as well as that of certain neurogenic genes, whereas it suppressed that of certain astrocytic genes in the embryonic neocortex. Importantly, our data indicated that the AT-hook 1 domain is necessary for both these activating and repressive functions of HMGA2 in neocortical NPCs. Chromatin condensation is often associated with gene repression, but it may also promote gene activation such as by facilitating enhancer-promoter interactions, as in the case of LLPS induced by the mediator complex[57]. The mechanisms by which HMGA2 differentially regulates gene activation and repression (in an AT-hook 1-dependent manner) have remained unclear. Posttranslational modifications as well as cofactors of HMGA proteins may contribute to their differential targeting and functions. For example, given that we detected the DNA repair protein MDC1 as an HMGA2 binding protein in neocortical cells and that DNA repair has been shown to play a role in HMGA-induced gene activation in MLE-12 cells[31], HMGA proteins may activate gene expression only when they interact with MDC1. With regard to their role in gene repression, it is possible that HMGA proteins cooperate with the repressive histone mark H3K9me3 (trimethylated Lys $^9$ of H3), which interacts with HMGA1 in human fibroblasts, in a locus- and context-dependent manner. It will be of interest to examine the relation between LLPS-mediated droplet formation induced by HMGA2 and that induced either by gene-activating complexes (such as Mediator and DNA repair proteins) or by gene-repressive complexes (such as H3K9me3- and HP1-dependent heterochromatin) within the nucleus.

Similar to major chromatin condensation factors such as H1, HP1, MeCP2, and PcG proteins, HMGA proteins may also play pivotal roles in the regulation of higher-order chromatin architecture and gene expression in various contexts and in combination with other epigenetic factors. In particular, given their essential roles in stem cell regulation in various tissues as well as in cellular senescence, HMGA proteins may confer a configuration of chromatin structure specific to stem cells or senescent cells. Further investigation of HMGA-mediated chromatin condensation should provide insight into not only normal tissue development but also HMGA-dependent tumorigenesis and senescence-related tissue impairment, and thereby contribute to cancer therapy and regenerative medicine.

## Methods

### Immunoprecipitation

Cells were dissociated from the E11.5 mouse neocortex with the use of Nerve Dispersion Solution (Wako) and stored at −80 °C, and Neuro2A cells overexpressing mouse HMGA2 were collected 1 day after the onset of transfection. A chromatin-bound fraction was prepared from the cells as previously described[58]. In brief, the cells were suspended and homogenized in a low-salt extraction buffer (20 mM Tris-HCl [pH 7.5], 100 mM KCl, 0.4 mM EDTA, 0.1% Triton X-100, 10% glycerol, 1 mM β-mercaptoethanol) supplemented with cOmplete Protease Inhibitor Cocktail (Roche), and the homogenate was centrifuged at $100,000 \times g$ for 30 min at 4 °C. A solubilized chromatin fraction was then isolated from the resulting pellet by suspension in a high-salt extraction buffer (20 mM HEPES-KOH [pH 7.0], 400 mM KCl, 5 mM MgCl$_2$, 0.1% Tween-20, 10% glycerol, 1 mM β-mercaptoethanol) supplemented with the protease inhibitor cocktail followed by centrifugation at $10,000 \times g$ for 30 min at 4 °C.

For immunoprecipitation of endogenous HMGA2, rabbit antibodies to HMGA2 (Cell Signaling, #8179) or control rabbit immunoglobulin G were cross-linked to protein A–Dynabeads (Thermo Fisher) with the use of Dess-Martin periodinane (Sigma). The antibody-cross-linked beads were then added to the solubilized chromatin fraction and incubated for more than 60 min at 4 °C, after which the beads were isolated by centrifugation and washed with the high-salt extraction buffer. The bead-bound proteins were then eluted with an

elution buffer (100 mM glycine-HCl [pH 2.5], 150 mM NaCl), and the eluate was neutralized with 1 M Tris-HCl (pH 8.0).

## MS analysis

Immunoprecipitated proteins were fractionated by SDS-PAGE on a 4–12% NuPAGE gel (Thermo Fisher) and stained with SimplyBlue (Thermo Fisher) for in-gel digestion. Portions of the gel containing protein bands were excised and cut into ~1-mm pieces, and the proteins in the gel pieces were reduced with dithiothreitol (Thermo Fisher), alkylated with iodoacetamide (Thermo Fisher), and digested overnight at 37 °C with trypsin and lysyl endopeptidase (Promega) in 40 mM $NH_4HCO_3$ (pH 8.0). The resultant peptides were analyzed with an Advance Ultra High Performance Liquid Chromatography system (AMR/Michrom Bioscience) coupled to a Q Exactive mass spectrometer (Thermo Fisher). The raw MS data were processed with Xcalibur (Thermo Fisher), and the liquid chromatography (LC)−MS results were then checked against the NCBI nonredundant protein/translated nucleotide database (restricted to *Mus musculus*) with the use of Proteome Discoverer version 1.4 (Thermo Fisher) and the Mascot search engine version 2.5 (Matrix Science). A decoy database composed of either randomized or reversed sequences in the target database was used for estimation of the FDR, and false positives were evaluated with the Percolator algorithm. Search results were filtered relative to a 1% global FDR for a high-confidence level.

## Preparation of recombinant HMGA2 proteins

Glutathione S-transferase (GST)- or hexahistidine-tagged recombinant human HMGA2, its deletion mutants and PUB1_IDR-HMGA2 hook1 del mutant were prepared as described previously[59]. In brief, DNA fragments encoding full-length or mutant forms of HMGA2 (N del: aa 1, 26–109, Hook1 del: aa 1–25, 35–109, Hook2 del: aa 1–45, 55–109, Hook3 del: aa 1–73, 83–109, C del: aa 1–94, Pub1_IDR-HMGA2 hook1 del: Pub1 aa 1, 243–327, HMGA2 aa 2–25, 35–109) were inserted into pET15b or pGEX-6P-1 vectors, which were then introduced into the *Escherichia coli* BL21 (DE3) codon plus RIL strain (Stratagene). Recombinant protein expression was induced by exposure of the bacterial cells to isopropyl-β-D-thiogalactopyranoside. The cells were subsequently lysed, GST-tagged HMGA2 was purified with the use of glutathione−Sepharose 4B beads (GE Healthcare), the GST tag was removed with PreScission protease, and the released HMGA2 fragment was purified by chromatography on a MonoS column (GE Healthcare) and stored at −80 °C. Hexahistidine-tagged HMGA2 was purified with the use of Ni-NTA beads (Qiagen) and stored at −80 °C.

## Preparation of histones and mononucleosomes

The human linker histone H1.2 was produced in and purified from bacterial cells as described previously[60]. Mononucleosomes were reconstituted with 145 bp or 193 bp of the Widom 601 sequence[61,62] by the salt-dialysis method as described previously[63]. The resulting nucleosomes were purified by native PAGE with the use of a Prep Cell model 491 apparatus (Bio-Rad).

## Pull-down assay

As a negative control, His-HMGA2 was treated with PreScission protease for 30 min at 16 °C. His-HMGA2 or PreScission-treated His-HMGA2 (1.7 μM) was incubated for 30 min at 4 °C with Ni-NTA beads (Qiagen) in 29 μl of a binding assay buffer (20 mM Tris-HCl [pH 8.0], 150 mM NaCl, 0.1% Nonidet-P40, 20 mM imidazole, 10% glycerol). The beads were then washed twice with the binding assay buffer and resuspended again in 30 μl of the same buffer. Mononucleosomes (5.0 μM) and recombinant H1.2 (1.0 μM) were added to the resuspended beads, and the mixture (final volume of 45 μl) was incubated for 30 min at 4 °C. The beads were washed twice with the binding assay buffer, resuspended in 10 μl of SDS sample buffer (100 mM Tris-HCl [pH 6.8], 4% SDS, 20% glycerol, 0.2% bromophenol blue, 5%

β-mercaptoethanol), incubated for 2 min at 98 °C, and subjected to SDS-PAGE and CBB staining.

## EMSA

Purified mononucleosomes (0.1 μM) were incubated for 30 min at 37 °C with recombinant H1.2 (0.7 μM) or HMGA2 (0, 0.15, 0.25, 0.35, 0.45 μM) in 10 μl of a solution containing 36 mM Tris-HCl (pH 7.5–8.0), 60 mM NaCl, 7% glycerol, 1.2 mM dithiothreitol, 1.2 mM β-mercaptoethanol, and bovine serum albumin (BSA, 5 μg/ml). The samples were analyzed by native PAGE, and the gel was stained with ethidium bromide and imaged with an LAS4000 image analyzer (GE Healthcare) or Amersham Imager 680 QC (GE Healthcare).

## Polynucleosome preparation

Polynucleosomes were prepared essentially as described previously[64]. Plasmid DNA containing the 12 tandem repeats of the Widom 601 sequence (repeat size of 208 bp) was isolated by EcoRV digestion as previously described[65], precipitated with polyethylene glycol, and dissolved in a solution containing 10 mM Tris-HCl (pH 8.0) and 0.1 mM EDTA (TE[10/0.1]). The dissolved DNA was further purified by DEAE chromatography with a TSKgel DEAE-5PW column (Tosoh), precipitated with ethanol, and dissolved in TE[10/0.1]. Polynucleosomes were then reconstituted with the dissolved DNA fragment and histone octamers by the salt-dialysis method.

## Self-association assay of nucleosomal arrays

Polynucleosomes (final concentration of 0.24 μM) were incubated for 15 min at room temperature with recombinant H1.2 (final concentration of 0.72 μM) or HMGA2 (final concentration of 0.48 μM) in 7.5 μl of a binding buffer (5% glycerol, BSA [0.15 mg/ml], 0.1 mM EDTA, 10 mM Tris-HCl [pH 8.0], 50 mM NaCl). Portions (10 μl) of the binding solution were then transferred to tubes containing 10 μl of twice the desired final concentration (0–4 mM) of $MgCl_2$ in the same buffer, and the mixtures were incubated for 10 min at room temperature. The samples were centrifuged at 12,500 × $g$ for 10 min at 4 °C, and 5 μl of the resulting supernatant were mixed with 5 μl of an SDS loading buffer containing 25% glycerol, 0.25% SDS, 20 mM Tris-HCl (pH 8.0), 80 mM EDTA, and proteinase K (0.5 mg/ml, Roche). The released DNA fragments were then analyzed by 1% agarose gel electrophoresis, ethidium bromide staining, and image analysis with ImageJ (U.S. National Institutes of Health [NIH]).

## AFM

Polynucleosomes (150 nM) and recombinant HMGA2 (1 μM) were incubated together for 10 min at 37 °C in 100 μl of a solution containing 28 mM Tris-HCl (pH 7.5–8.0), 20 mM NaCl, 3% glycerol, 1.2 mM dithiothreitol, 0.4 mM β-mercaptoethanol, and bovine serum albumin (BSA, 5 μg/ml), after which glutaraldehyde was added to a final concentration of 0.1%. The mixture was maintained for 30 min on ice, dialyzed overnight at 4 °C with a dialysis buffer consisting of 10 mM Tris-HCl (pH 7.5), 0.1 mM EDTA, and 15 mM NaCl, and stored at 4 °C. The mica substrate (Alliance Biosystems) for AFM was coated with 0.1% poly-L-ornithine (Fujifilm) for 2 min at room temperature and then washed twice with deionized water before the addition of the polynucleosomes (~6 nM) and incubation for 5 min at room temperature. The substrate was then washed twice with the dialysis buffer and observed by AFM with a NanoWizard IIR instrument (JPK) and BL-AC40TS-C2 Bio Lever Mini cantilever (Olympus). Images were acquired in QI mode with the sample in dialysis buffer. ImageJ (NIH) was used for image analysis.

## DNase I sensitivity assay

Polynucleosomes (final concentration of 1.0 μM) were incubated for 30 min at 37 °C with recombinant H1.2 (final concentration of 3.0 μM) or HMGA2 (final concentration of 0, 2.0, 4.0 μM) in 40 μl of a binding

buffer (5% glycerol, BSA [0.15 mg/ml], 0.1 mM EDTA, 10 mM Tris-HCl [pH 8.0], 50 mM NaCl). 32 μl of samples were incubated with 8 μl of DNase I (0.07U, Takara) in 20 mM Tris-HCl (pH 7.5) buffer containing 50 mM NaCl for 0, 3 or 9 min at 37 °C. The reaction was terminated by the addition of 5 μl of a deproteinization solution (20 mM Tris-HCl [pH 8.0], 25% glycerol, 80 mM EDTA, proteinase K [3 mg/ml, Roche], 0.25% SDS) to 10 μl of the reaction mixture. The released DNA fragments were analyzed by 1.2% agarose gel electrophoresis and ethidium bromide staining.

## MNase sensitivity assay
Mononucleosomes (final concentration of 0.2 μM) were incubated for 30 min at 37 °C with recombinant Nap1 (final concentration of 0.3 μM), H1.2 (final concentration of 1.2 μM) or HMGA2 (final concentration of 0.6 μM) in 50 μl of 12 mM Tris-HCl (pH 7.5−8.0), 55 mM NaCl, 5.5% glycerol, 0.2 mM dithiothreitol, 0.8 mM β-mercaptoethanol, 0.05 mM EDTA and 0.01 mM phenylmethylsulfonyl fluoride. 6.6 μl of the samples were incubated with MNase (70 mU/μl, Takara) for 0, 3, 9, or 15 min at 37 °C in 3.3 μl of a solution containing 20 mM Tris-HCl (pH 8.0), 2.5 mM CaCl$_2$, 5 mM NaCl, and 1 mM dithiothreitol. The reaction was terminated by the addition of 5 μl of a deproteinization solution (200 mM Tris-HCl (pH 8.8), 80 mM EDTA, proteinase K [0.5 mg/ml, Roche], 0.25% SDS) to 5 μl of the reaction mixture. The released DNA fragments were analyzed by native PAGE on an 8% gel and ethidium bromide staining.

## Hmga1/2 and Hmga2 cKO as well as Hmga2-EGFP knock-in mice
*Hmga1*flox/flox (*Hmga1*fl/fl) or *Hmga2*flox/flox (*Hmga2*fl/fl) mice[45,66] were crossed with *Sox1-Cre* transgenic mice[44] to generate corresponding cKO animals. Jcl:ICR (CLEA Japan) or Slc:ICR (SLC Japan) mice were studied as wild-type animals. All mice were maintained in a temperature- and relative humidity-controlled environment (23° ± 3 °C and 50 ± 15%, respectively) with a normal 12-h-light, 12-h-dark cycle. The mice were housed two to six per sterile cage (Innocage, Innovive; or Micro Barrier Systems) with chips (Palsoft, Oriental Yeast; or PaperClean, SLC Japan), and with irradiated food (CE-2, CLEA Japan) and filtered water available ad libitum. Mouse embryos were isolated at various ages, with E0.5 being considered the time of vaginal plug appearance. All animals were maintained and studied according to protocols approved by the Animal Care and Use Committee of The University of Tokyo.

For generation of a targeting vector for the establishment of *Hmga2-EGFP* knock-in mice, homology fragments of the *Hmga2* gene were amplified by genomic PCR from the RPCI-23 BAC library (*Mus musculus*, strain C57BL/6J). A 2.4-kb 5′ arm containing a portion of the open reading frame immediately before the stop codon of *Hmga2* was cloned immediately before the start codon of the EGFP gene in the pEGFP-IRES-neo3 vector[67], and a 4.1-kb 3′ arm containing the 3′ untranslated region immediately downstream of the stop codon of *Hmga2* was cloned into the 3′ multiple-cloning site of the vector. The linearized targeting vector was introduced into wild-type TT2-KTPU8 F1 mouse ES cells by electroporation, and the cells were then subjected to selection with G418. Homologous recombination was identified by Southern blot screening of G418-resistant colonies. The gene-targeted ES cells were then aggregated with morulae of ICR mice. The aggregated embryos were transferred to pseudopregnant females and allowed to develop to term. The chimeric offspring were bred with wild-type C57BL/6 mice, and the resulting pups were screened for the presence of the heterozygous targeted allele. The genotype of the mice was determined by Southern blot analysis and PCR analysis of genomic DNA isolated from the tail or ear. Heterozygous mice were intercrossed to obtain homozygous mice. Isolated mouse embryos were fixed for 4 h with 4% paraformaldehyde (Merck) in phosphate-buffered saline (PBS), washed with PBS, exposed consecutively to 15% and 30% sucrose in PBS, embedded in O.C.T. compound (Sakura Finetek Japan) at −80 °C, stained with Hoechst 33342 and imaged by a Zeiss LSM 880 microscope.

## FACS
The neocortex was dissected and subjected to enzymatic digestion with Nerve Dispersion Solution (Wako). The dissociated single cells were isolated and incubated for 10 min at room temperature with PBS containing allophycocyanin-conjugated antibodies to CD133 (1:200 dilution, BioLegend, 372805) (for E11.5 and E12.5 samples), or phycoerythrin- and Cy7-conjugated antibodies to CD133 (1:200 dilution, BioLegend, 141210) and allophycocyanin-conjugated antibodies to CD24 (1:200 dilution, BioLegend, 101814) (for P1 samples). Cells were directly subjected to fluorescence-activated cell sorting (FACS) with a FACS Aria instrument (Becton Dickinson). Debris and aggregated cells were removed by gating on the basis of forward and side scatter. NPCs from E11.5 and E12.5 samples are collected on the basis of the presence of NPC marker CD133 (also known as prominin)[68]. NPCs from P1 samples are collected on the basis of the presence of NPC marker CD133 and the absence of the neuronal marker CD24[69].

## ChIP-seq analysis
ChIP analysis was performed as previously described[15] with antibodies to HMGA2 (#8179, Cell Signaling; or in-house[16]) and to H3K27me3 (MBL). Two million cells directly isolated from the neocortex of embryos at E11.5 were used for ChIP-seq of HMGA2, and NPCs isolated as CD133high cells by FACS from the neocortex of embryos at E12.5 were used for ChIP-seq of H3K27me3. Template preparation was performed with the use of an Illumina TruSeq ChIP Sample Preparation Kit, and deep sequencing was performed on the Illumina HiSeq platform to obtain 36-base single-end reads. Sequences were mapped to the reference mouse genome (mm10) with the use of Bowtie software[70]. Only uniquely mapped reads without blacklist regions determined by the ENCODE project[71] were accepted. Peaks for HMGA2 were called with the use of F-seq software[72], and ngsplot[73] was adopted for clustering and heat map construction. Correlation analysis was conducted by deeptools[74]. RepEnrich[75] was used for repeat analysis.

## DNase I-seq analysis
NPCs (6.0 × 10$^4$) isolated as CD133high cells by FACS from the neocortex of mouse embryos at E12.5 were suspended in 200 μl of Nuclear Buffer A (85 mM KCl, 5.5% sucrose, 10 mM Tris-HCl [pH 7.5], 0.5 mM spermidine, 0.2 mM EDTA). An equal volume of Nuclear Buffer B (85 mM KCl, 5.5% sucrose, 0.1% Nonidet P-40, 10 mM Tris-HCl [pH 7.5], 0.5 mM spermidine, 0.2 mM EDTA, 0.2% BSA) was then added, and the mixture was agitated gently, on ice for 3 min, and centrifuged at 600 × g for 10 min at 4 °C. The resulting nuclear pellet was resuspended in 50 μl of Nuclear Buffer R (85 mM KCl, 5.5% sucrose, 10 mM Tris-HCl [pH 7.5], 3 mM MgCl$_2$, 1.5 mM CaCl$_2$), DNase I (K1901BA, Takara) was added to a final concentration of 2 U/ml, and the mixture was incubated for 10 min at 37 °C. 250 μl of Lysis buffer (50 mM Tris-HCl [pH 8.0], 1% SDS, 10 mM EDTA) was added, and DNA was purified by phenol-chloroform extraction. Purified DNA was subjected to agarose gel electrophoresis, and portions of the gel containing DNA fragments of <1 kbp generated by DNase I were excised, and the DNA fragments were extracted and purified with the use of a FastGene Gel/PCR Extraction Kit (Nippon Genetics). Template preparation was performed with the use of a TruSeq ChIP Sample Prep Kit-set A/B (Illumina), and deep sequencing was performed on the Illumina HiSeq2500 platform. The obtained reads were mapped to the mouse mm10 genome with the use of Bowtie[70], with a maximum of only two mismatches allowed. Only sequence reads that specifically matched only one location in the mm10 genome sequence were used for subsequent analysis. Visualization was performed with ngsplot[73].

## ATAC-seq analysis
ATAC-seq analysis was performed as described previously[76]. NPCs (5 × 10$^4$) isolated as CD133high cells by FACS from the neocortex of mouse embryos at E11.5 were lysed by a lysis buffer (10 mM Tris-HCl [pH 7.4], 10 mM NaCl, 3 mM MgCl$_2$, 0.1% IGEPAL CA-630) and

treated with Tn5 with the use of a Nextera DNA Library Preparation Kit (Illumina). DNA was amplified with the use of NEBNext High-Fidelity 2×PCR Master Mix (New England BioLabs) and purified with AMPure XP (Beckman Coulter). The quality of the purified DNA was checked with a BioAnalyzer (Agilent), and the DNA was sequenced with an Illumina HiSeq 3000 system. Sequences were mapped to the reference mouse genome (mm10) with Bowtie software[70]. Only uniquely mapped reads without blacklist regions determined by the ENCODE project[71] were accepted. Visualization was performed with ngsplot[73].

### Hi-C analysis

Hi-C analysis was performed as previously described previously[77]. Briefly, the neocortex of mouse embryos at E11.5 were dissociated with Nerve Dispersion Solution (Wako), and cells ($2 \times 10^6$) were fixed for 10 min at room temperature with freshly prepared 1% formaldehyde in PBS. The reaction was quenched for 5 min by the addition of 2.0 M glycine (final concentration of 200 mM), followed by lysis, restriction digest, marking of DNA ends, proximity ligation, cross-link reversal, DNA shearing and size selection[77]. Obtained DNA was pulled down by Dynabeads MyOne Streptavidin T1 beads (Life Technologies) and subjected to library preparation with KAPA Hyper Prep Kit (Nippon Genetics). The DNA was sequenced with an Illumina HiSeq X system. The obtained reads were mapped with the use of Hi-C juicer[78]. Compartments were defined by "hicPCA" in HiCExplorer[79] and distinguished by positivity or negativity for the first principal component, with the compartment with the higher gene density being the A compartment and the compartment with the lower gene density being the B compartment.

### FRAP analysis

For in cellulo FRAP, the neocortex of E11.5 *Hmga2-EGFP* mice was dissected and dispersed with Nerve Dispersion Solution (Wako) and the released cells ($2.5 \times 10^5$) were seeded in 35-mm glass-bottom dishes (Iwaki), washed twice with ethanol and once with water, and then cultured overnight under 5% $CO_2$ at 37 °C in 2.5 ml of Dulbecco's modified Eagle's medium (DMEM)–F12 (Gibco) supplemented with recombinant human fibroblast growth factor 2 (20 ng/ml, Invitrogen) and B27 (final concentration of 2%, Invitrogen). After the addition of 50 µl of 1 M HEPES-NaOH (pH 7.2–7.5, Thermo Fisher), the cells were maintained for 3 min at 37 °C and then subjected to FRAP analysis with a confocal microscope (Leica SP5). Eight photographs were acquired before bleaching, and the cells were observed for 40 s after bleaching. The sizes of the photographed and bleached areas were maintained constant. Sufficient bleaching was confirmed by bleaching fixed samples at the same intensity. Fluorescence intensity in images was calculated with ImageJ (NIH). The recovery curve for normalized fluorescence intensity was calculated after subtraction of background fluorescence intensity.

For in vitro FRAP analysis, recombinant HMGA2 was reconstituted by dialysis in PBS adjusted to pH 8.5 and was incubated for 4 h at room temperature with 2.9 µM ATTO 647 NHS ester (Sigma). The labeled HMGA2 was further purified by filtration with an Amicon 10 K device (Millipore) in the buffer containing 20 mM Tris-HCl (pH 8.0), 100 mM NaCl, 10% glycerol and 2 mM β-mercaptoethanol. Labeled HMGA2 and unlabeled HMGA2 were mixed at a ratio of 3:7 for FRAP analysis in the droplet formation assay described in the next section. Images were collected with a Zeiss LSM 880 microscope every 0.5 s, and the postbleach image was acquired at 55 s after photobleaching. Combined images were processed with ZEN software, and fluorescence intensity values in images were calculated with ImageJ (NIH).

### In vitro droplet formation assay

The in vitro LLPS assay of full-length or mutant HMGA2 was performed for 10 min at 37 °C in a solution containing 12 mM Tris-HCl

(pH 7.5–8.0), 0.2 mM dithiothreitol, 0.4 mM β-mercaptoethanol, and 5% glycerol and with various concentrations of HMGA2 and NaCl in the absence or presence of mononucleosomes (800 nM). Assay of PUB1$_{IDR}$-HMGA2 hook1 del mutant was conducted in a solution containing 16 mM Tris-HCl (pH 7.5–8.0), 0.2 mM dithiothreitol, 300 mM NaCl, 1.2 mM β-mercaptoethanol, and 7% glycerol. Fluorescence and DIC images were acquired with a confocal microscope (Leica TCS-SP5 or Zeiss LSM 880) and were processed and analyzed with ImageJ (NIH). Turbidity was determined by measurement of $OD_{400}$ with a Nano-drop ND-1000 instrument (Thermo Fisher). Polyethylene glycol 8000 (Sigma) was added to a final concentration of 10% for FRAP analysis.

### RNA-seq analysis

RNA-seq was performed as previously described[80]. NPCs were isolated by FACS either as CD133$^{high}$ cells from the neocortex of unmanipulated embryos at E12.5 ($5 \times 10^4$ cells) or as CD133$^{high}$CD24$^{low}$ cells from the neocortex of electroporated embryos at P1 ($>1 \times 10^4$ cells). Total RNA was isolated from the cells for library construction. Template preparation was performed with the use of a SMART-Seq Stranded Kit (Takara), and deep sequencing was performed on the Illumina HiSeq2000 platform to obtain 150-bp paired-end reads. About 20 million sequences were obtained and were mapped to the reference mouse genome (mm10) with the use of Hisat2[81]. Only uniquely mapped and "deduplicated" reads with no base mismatch were used. Gene expression was quantitated as TPM (transcripts per million) on the basis of RefSeq gene models (mm10) with the use of featureCounts[82]. DEGs were identified with edgeR of the R package[83] as genes whose $p$ values were <0.05. GO analysis was performed with DAVID software[84]. The read counts between independent experiments were normalized by RUVSeq[85].

### Cell culture and transfection

IMR90 cells and Neuro2A cells were cultured under 5% $CO_2$ at 37 °C in Eagle's minimum essential medium (E-MEM, Fujifilm) or DMEM (Fujifilm), respectively, each supplemented with 10% fetal bovine serum (JRH Bioscience or Gibco) and 1% penicillin-streptomycin (Invitrogen). Cells were seeded in 24-well plates ($5 \times 10^3$ cells per well) and cultured for 1 day before transfection with the use of polyethyleneimine (Polysciences), GeneJuice (Millipore) or Lipofectamine LTX (Thermo Fisher) reagents. Neuro2A cells were transfected for 1 day with pCAG-Hmga2[8] encoding mouse HMGA2 before IP-MS analysis, and IMR90 cells were transfected for 3 days with cUX encoding GFP-tagged human HMGA2 (full length or hook1 del mutant) (modified from cUX-IRES-EGFP[86]) before fluorescence microscopic imaging as described in the next section.

### Cell imaging

Cells were fixed with 4% paraformaldehyde (Merck) for 10 min at room temperature, washed twice with PBS, incubated for 10 min at room temperature with 0.2% Triton X-100 in PBS, washed twice with PBS, and incubated at room temperature first for 30 min in PBS containing 3% BSA and then for 10 min with Hoechst 33342 in the same solution. They were finally observed with a laser confocal microscope (Leica TCS-SP5 or Zeiss LSM 880).

### FRET analysis

Mononucleosomes were reconstituted with 193 bp of the Widom 601 sequence labeled with Cy3 (FASMAC) and Cy5 (FASMAC) of its terminus. Five microliters of mononucleosomes (20 nM) were mixed with 50 µl of recombinant HMGA2 (0 to 80 µM), 25 µl of Tris-EDTA buffer, and 1 µl of 3% Nonidet-P40, and the volume of the mixture was increased to 100 µl by the addition of water. The fluorescence was measured with a scanner (Amersham Typhoon). Data were analyzed with ImageJ (NIH).

## In utero electroporation

Plasmid DNA (pCAG2IG, which expresses GFP alone, or the same vector encoding full-length HMGA2, hook1 del HMGA2 mutant or PUB1$_{IDR}$-HMGA2 hook1 del mutant was introduced into NPCs of the developing mouse embryonic neocortex as previously described[87,88]. In brief, plasmid DNA was injected into the lateral ventricle at the indicated developmental stages, electrodes were positioned at the flanking ventricular regions, and four 50-ms pulses of 32 to 35 V were applied at intervals of 950 ms with the use of an electroporator (CUY21E, Tokiwa Science). The uterine horn was returned to the abdominal cavity to allow continued development of the embryos.

## Immunohistofluorescence analysis

For immunohistochemical staining of electroporated brain sections, mice were transcardially perfused with ice-cold 4% paraformaldehyde (Merck) in PBS. The brain was then removed, exposed to the same fixative for 4 h at 4 °C, equilibrated with 30% sucrose in PBS, embedded in O.C.T. compound (Tissue TEK), and frozen. Coronal cryosections (thickness of 12 μm) were exposed to Tris-buffered saline containing 0.1% Triton X-100 and 3% BSA (blocking buffer) for 1 h at room temperature, incubated first overnight at 4 °C with primary antibodies in blocking buffer and then for 1 h at room temperature with Alexa Fluor-conjugated secondary antibodies (Thermo Fisher) and Hoechst 33342 (1:10,000 dilution, Molecular Probes) in blocking buffer, and mounted in Mowiol (Calbiochem). Fluorescence images were obtained with a laser confocal microscope (Leica TCS-SP5 or Zeiss LSM 880) and were processed with the use of LAS AF (Leica), ZEN (Zeiss), and ImageJ (NIH) software. For HMGA2-GFP mice, sections were subjected to antigen retrieval with Target Retrieval Solution (Agilent) at 105 °C for 10 min before blocking. Primary antibodies included chicken anti-GFP (1:1000 dilution, Abcam ab13970), rat anti-GFP (1:1000 dilution, Nacalai Tesque GF090R), rabbit anti-GFP (1:1000 dilution, MBL 598), rabbit anti-Sox2 (1:200 dilution, Cell Signaling 3728), chicken anti-Tbr2 (1:500 dilution, Millipore AB15894), goat anti-NeuroD1 (1:100 dilution, Santa Cruz Biotechnology sc-1084), rabbit anti-Ki67 (1:500 dilution, Abcam ab16667), HP1α (1:500 dilution, CST 2616S), and H3K9me3 (1:800 dilution, Thermo Fisher MABI 0319).

## Assay of thermal stability of nucleosomes

The thermal stability assay was performed as previously described[89]. The fluorescence signal from SYPRO Orange, which binds hydrophobically to thermally denatured histones released from nucleosomes, was monitored. Mononucleosomes (equivalent to a final DNA concentration of 0.25 μg/μl) were incubated with HMGA2 (final concentration of 0, 1.1, 3.4 μg/μl) in 18 μl of a solution containing 18 mM Tris·HCl (pH 7.5), 0.9 mM dithiothreitol, 60 mM NaCl, and SYPRO Orange (Thermo Fisher). The fluorescence signal was detected with a StepOnePlus Real-Time PCR unit (Applied Biosystems). A temperature gradient (26° to 95 °C in steps of 1 °C/min) was applied. Fluorescence data were normalized to percentage values according to $(F(T) - F(26 °C))/(F(95 °C) - F(26 °C))$, where $F(T)$, $F(26 °C)$, and $F(95 °C)$ are the fluorescence values at a particular temperature, 26 °C, and 95 °C, respectively.

## Statistical analysis

Data are presented as means ± s.d. and were compared between two groups with the two-tailed Student's $t$ test or the Mann–Whitney U test, or among three or more groups by one-way analysis of variance (ANOVA) followed by Dunnett's test or Tukey's multiple comparison test using GraphPad Prism version 8 for Windows.

## Reporting summary

Further information on research design is available in the Nature Portfolio Reporting Summary linked to this article.

## Data availability

The data and codes that support this study are available within the manuscript, in the associated source data file and from the corresponding authors upon request. The sequence data have been deposited in the DNA Data Bank of Japan (DDBJ) Sequence Read Archive under the following accession codes: DRA008363, DRA008364, DRA010294, DRA010295 (HMGA2 ChIP-seq), DRA015284 (DNase I-seq), DRA015285 (ATAC-seq), DRA015234 (Hi-C), DRA017008 (H3K27me3 ChIP-seq) and DRA015260, DRA015261, DRA015262, DRA016538 (RNA-seq). Supplementary files have also been deposited in the DDBJ Genomics Expression Archives under the accession code E-GEAD-571, E-GEAD-572, E-GEAD-573, E-GEAD-574, E-GEAD-575, E-GEAD-576, E-GEAD-577, E-GEAD-578, E-GEAD-581, E-GEAD-582, E-GEAD-624 (https://ddbj.nig.ac.jp/public/ddbj_database/gea/experiment/E-GEAD-000/). Source data are provided with this paper.

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

## Acknowledgements

We thank S. Nishikawa (RIKEN) for providing *Sox1-Cre* mice; K. Imamura, T. Horiuchi, S. Sugano, and Y. Suzuki (The University of Tokyo) for performing high-throughput sequencing analysis; H. Ichijo (The University of Tokyo) for FUS encoding plasmids; K. Tsukamoto (Bruker Japan) for the help of AFM manipulation; the One-Stop Sharing Facility Center for Future Drug Discoveries (The University of Tokyo) for FACS; the Joint Usage/Research Center for Developmental Medicine, IMEG, Kumamoto University for IP-MS analysis; Y. Maeda, R. Nagayoshi, Y. Kakeya, R. Ogawara, and Y. Kuroda for technical assistance; and members of the Gotoh laboratory for discussion. This research was supported by AMED-CREST and AMED-PRIME of the Japan Agency for Medical Research and Development (JP22gm1310004, JP22gm6110021), SECOM Science and Technology Foundation SECOM Science and Technology Foundation (for Y.K.), Platform Project for Supporting Drug Discovery and Life Science Research from AMED JP21am0101076 and (for H.K.), Research Support Project for Life Science and Drug Discovery from AMED JP22ama121009 (for H.K.), Japan Science and Technology Agency ERATO JPMJER1901 (for H.K.) and by KAKENHI grants from the Ministry of Education, Culture, Sports, Science, and Technology of Japan and the Japan Society for the Promotion of Science (JP21J14115 for N.K.; JP22K15033 for T.K.; 16H06279, 20H03179, 21H00242, and 22H04687 for Y.K.; 20K07589 for S.W.; JP20H00449, JP18H05534 for H.K.; JP22H00431, JP16H06279, and JP22H04925 for Y.G.).

## Author contributions

N.K.: conception and design, collection and assembly of data, data analysis and interpretation, and manuscript writing. T.K.: study conception, supervision and interpretation of biochemical experiments. Y.K.: study conception, design, collection, assembly, analysis and

interpretation of data, and supervision. R.H., K.E., and L.F.: technical assistance. S.W. and M.N.: generation of Hmga2-EGFP mice. Y.S.: high-throughput sequencing analysis. K.I.: collection, analysis, and interpretation of IP-MS experiments. H.K.: data interpretation and study supervision. Y.G.: study conception and design, data interpretation, financial and administrative support, supervision, and manuscript writing. All authors: revision and final approval of the manuscript.

## Competing interests

The authors declare no competing interests.
