## [Peer Review File · Nature Communications]

HMGA2 directly mediates chromatin condensation in association with neuronal fate regulationREVIEWER COMMENTS

Reviewer #1 (Remarks to the Author):

This is an interesting paper on the properties and role of the HMGA2 protein in neuronal progenitor cells. The authors use a large variety of techniques to this end and provide a large body of data supporting HMGA2 being involved in chromatin condensation and phase separation as well as being vital for neurogenesis. The data is compelling; however, there are parts where the data appears to be overinterpreted and raises limited concerns. Overall, this is a strong paper that needs some corrections/adjustments.

A broad range of biochemical approaches was used to explore the ability of HMGA2 to bind nucleosomes, condense polynucleosomes including AFM, form droplets in vitro and FRET based assays for folding linker DNA. They used His-tagged HMGA2 to show it binds to nucleosomes and does not directly bind to the linker histone H1.2. However, it is not clear if HMGA2 is directly binding the nucleosome or is merely binding the linker DNA present at both sides of the nucleosome and is merely being mediated by DNA. The authors should have performed these experiments with only the core nucleosome to be able to conclude direct binding to nucleosomes. The polynucleosome experiments rely on the concentration of MgCl₂ required to aggregate polynucleosomes with or without HMGA2 (and H1.2) added. The biological relevance of this assay for measuring chromatin condensation is somewhat of a question in the field particularly in terms of the amounts of the varied amounts of MgCl₂ required for aggregation, so I'm not confident as to the over value of these experiments. The AFM experiments are probably a more preferred way for showing HMGA2 promotes condensation. From these experiments the extent of condensation seems to be modest and was wondering if the authors could make a more quantitative statement about the degree of change observed. The MNase experiments with mononucleosomes and HMGA2 and H1.2 were insightful as to the binding of these two proteins, but the statement that when both H1.2 and HMGA2 are present the extent of protection is greater than with H1.2 alone was not readily evident from the data provided. I don't think this needs to be the case for the point the authors are trying to establish.

The HMGA2 ChIP-seq done in parallel with ATAC-seq and DNase I-seq in mouse embryonic neocortical cells was very impactful and supports that HMGA2 binding causes DNA to be less accessible and likely more compacted in vivo. The finding from the Hi-C data that HMGA2 has a preference to be localized in B-compartments or transcriptionally inactive regions is further support for this point. The EGFP tagged Hmga2 locus mice provided strong evidence for Hmga2 being localized to condensed chromatin and together all of this provided compelling evidence for the role of HMGA2 in forming heterochromatic regions in vivo. The engineered NPC used to conditionally knock out Hmga1 and Hmga2 was a further demonstration of their role in gene repression and again is an excellent set of data. The droplet assay provides evidence for HMGA2 with nucleosomes promoting phase separation and the corresponding formation of droplets. More could be done with these assays, but in this current format is sufficient for this paper.

When performing domain deletions of HMGA2 to determine what are the domains required for chromatin condensation, I was left with the question why the authors did not test the AT-hook 2 domain and instead only examined AT-hooks 1 and 3? The mutant HMGA2 proteins were only tested using the MgCl₂ aggregation assays which is probably an oversight, rather than the AFM assays for the reasons described above. The FRET assay used to assay the conformation of the linker DNA after binding of H1.2 and HMGA2 is potentially a powerful approach. I have not seen FRET data presented in this way and was looking for more quantitative analyses. The conclusion that HMGA2 restricts linker DNA movement is not clear or obvious given the extent of FRET change is not readily detected. The other *in vivo* assays showing the AT-hook is important for puncta formation and *in vitro* for droplet formation were good and support the overall conclusions of the paper. The experiments to test whether the AT-hook 1 is required for HMGA2 maintenance of neurogenic progenitors were particularly important experiments. The bar graph showing the quantitative differences between wild type and AT-hook deletion mutant however did not seem to agree with the immunofluorescence images shown in Figure 7. When comparing full length versus AT-hook deletion the quantitation indicates nearly a 10-fold decrease, but I can't visually see that in the image.

Up until the end of the paper the message of HMGA2 has all been about gene repression, but at the end RNA-seq analysis is done after over expression of different forms of HMGA in the embryonic neocortex and the authors focus on the activation of transcription by HMGA2. I was left wondering how does HMGA2 activate transcription, how does this relate to the rest of the paper and why is the activation properties of HMGA2 the key for maintenance of proliferating cells rather than repression and chromatin condensation.

Reviewer #2 (Remarks to the Author):

The study by Kuwayama et al. investigates the mechanism of HMGA2-mediated regulation of gene expression. The authors identify histone H1 as one of the interaction partners of HMGA2 in the developing mouse neocortex, and, using *in vitro* reconstitution assays, show that HMGA2 promotes chromatin condensation. Motivated by this finding, they analyze HMGA2 genomic localization in the neocortex by ChIP-seq and correlate binding with open chromatin (ATAC-seq, DNase I-seq) and A/B chromatin compartments (HiC). This analysis revealed reduced chromatin accessibility and preferential enrichment in B compartments, indicative of heterochromatin, for HMGA-bound sites. Based on *in vitro* data, the authors propose that HMGA2 has intrinsically disordered domains and forms droplets. Loss of HMGA1/2 or HMGA2 in neural progenitor cells lead to upregulation of HMGA2 bound genes. Using structural deletion mutants of HMGA2, AT-hook 1 is shown to be important for chromatin condensation. Finally, overexpression of HMGA2 in the developing mouse neocortex resulted in an increased number of progenitor cells, along with expression of neurogenic genes and reduction of astrogenic genes, which requires the AT-hook 1 or other unstructured domains to be present in HMGA2.

This work represents an elegant combination of *in vitro* biochemical assays and *in vivo* work

in the developing neocortex, where HMGA2 contributes to gene regulation. Understanding the molecular mechanisms of gene regulation is important in the context of development and disease. In this manuscript, the authors analyze intrinsic protein properties and functions *in vitro*, and then move on to examine the relevance of their findings in a biological system of great relevance, the developing brain. Overall, this study greatly advances our understanding of the function of abundant non-histone chromatin proteins.

While the experiments are overall well-controlled and support the conclusions put forward in the manuscript, there are some points that should be considered:

1. The chromatin data (CHIP-seq, ATAC-seq, DNase I-seq, HiC) in Figure 3 requires more in-depth analysis to allow assessment of the quality of the data (replicates, location of peaks in relation to genomic features, similarity of CHIP with the different HMGA2 antibodies). Example tracks and peaks should be presented for representative loci. What is the nature of the repressive chromatin (i.e., genes, centromeres, telomeres, repeats etc.)? Is there any correlation with other repressive modifications (H3K27me3, H3K9me3)? While accession numbers to deposited data sets were included in the manuscript, the data was not accessible to the reviewer.
2. In Figure 3F, a DNA stain would be helpful to appreciate the enrichment in the VZ. Are the cells on the right from tissue or cultured cells? If indeed VZ, it is surprising to see the low packaging of cells.
3. Does droplet formation dependent on protein concentration (Figure 4)? Is this in a similar range to other proteins shown to undergo LLPS?
4. For the RNA-seq expression analysis, it would be helpful to see the data presented as expression changes in HMGA2-bound regions. What fraction goes up/down? Which kind of genes?
6. In Figure 6F, is pericentric heterochromatin overall disturbed by expression of HMGA2 hook 1 del? In the DNA stain, the heterochromatin foci are not apparent.
7. For the *in utero* electroporation images in Figure 7, a DNA stain should be presented for all conditions to appreciate the effect on tissue architecture and integrity.
8. The authors propose that lack of the AT hook 1 in HMGA2 only affects the chromatin condensation function of HMGA2. However, given that AT hook domains in general are known to contribute to DNA binding, does the hook 1 deletion protein actually bind to chromatin in a similar way as the full-length HMGA2 protein? This should be tested by CHIP. If the hook 1 deletion protein does not bind to chromatin, this would be an alternative explanation for its reduced ability to induce HMGA2 phenotypes.
9. There is a typo in line 443 (McCP2).

Reviewer #3 (Remarks to the Author):

In this work, the authors studied 1) the interaction between HMGA2, H1 and nucleosomes; 2) the chromatin location and transcription regulation of HMGA2; 3) LLPS of HMGA2 and 4) HMGA2 maintaining neurogenic progenitors. They found some interesting results, including both HMGA2 and H1 interact with nucleosomes, HMGA2 localized to transposase 5– and DNase I– inaccessible chromatin regions, HMGA2 binding was mostly associated with gene repression, the AT-hook 1 domain was necessary for chromatin condensation by HMGA2 in vitro and in cellulo, and an HMGA2 mutant lacking this domain was defective in the ability to maintain neuronal progenitors in vivo. However, the relevance between these results has not been fully studied. I also have some concerns about some conclusions and experiment designations.

1. The first section title “HMGA2 interacts with linker histone H1” does not have enough supports in this work. The fact is that both HMGA2 and H1 interact with nucleosomes, which can explain all results in Fig. 1. The authors did not analyze the competition between HMGA2 and H1 by quantitative experiments, so it’s improper to explain the result with the statements that HMGA2 does not compete, but rather forms a complex, with H1 in cells (Page6 line140-141).

2. The authors analyzed HMGA2 ChIP-seq with ATAC-seq, DNase I-seq and AB compartments, draw the conclusion that HMGA2 localizes to heterochromatin in the mouse neocortex. I think it’s better to use more direct heterochromatin marker like ChIP-seq of HP1 and H3K9me3.

3. HMGA2 has three AT-hook domains with similar sequences, why did the authors only study AT-hook 1? Is it possible that three domains have same effects?

4. PUB1IDR-HMGA2 hook1 del and HMGA2 hook1 del-FUSIDR phenocopied full-length HMGA2. Can PUB1IDR-HMGA2 hook1 del and HMGA2 hook1 del-FUSIDR undergo LLPS in vitro with or without nucleosomes? Can they regulate the genes regulated by WT HMGA2?

5. HMGA2 cannot form LLPS droplet by itself, but PUB1IDR or FUSIDR undergo LLPS in vitro by themselves. Besides, PUB1IDR or FUSIDR are much longer than hook1 domain. I think such replacement experiments are not reasonable.

6. Page6 line148: missing left bracket.

7. In the chart of Fig. 1A. Anti-HMGA2-#1 perform consistently with known findings, as well as the conclusions made here that HMGA2 is associated with H1 and H2B. While, it’s confusing why the PSM MDC1 is “0” for the case of anti-HMGA2-#2, especially leading to inconsistent decisions about associations with H2B and MDC1. It’s just slightly higher of the PSM of H1 proteins for anti-HMGA2-#2. Comprehensively compared with all these interacting proteins listed in 1A that were IP-ed using the two antibodies, anti-HMGA2-#2 performed worse.

8. The authors should carefully arrange the different panels of Fig. 3f, and legend as well. For example, (f) Fluorescence microscopy of a coronal section of the Hmga2-EGFP mouse neocortex at E11.5 (top, scale bar = 200 μ m) as well as of a portion of the ventricular zone also showing Hoechst 33342 staining (bottom, scale bars = 5 μ m). What is the top or bottom panel referred to?

9. Legends of fig.4a and 4b are lack of scalebars and indicated time for individual image. The

line graph of FRAP is described that “Data are means \pm s.d. (n = 72 cells from three independent experiments)”. How many droplets and cells are calculated here?

10. Fig.4e and 4f. Actually, the optic resolution is far away qualified. Meanwell, the results of independent experiments should be supplied in sFigure. And also lack scalebar onto images. LLPS should be confirmed with a series of detail assays. It is very advised to construct phase diagrams by observing the droplet in the presence of a range of environmental conditions, typically varied protein concentrations and salt contents. The two phases start to separate in a switch-like manner. One should figure out the boundary.

11. Check order numbers in the legend of Fig. 6: e,f,g,f,g,i?

12. Fig. 7: What’s the plasmid introduced or expressed in control neocortex? Only GFP expressing plasmid?

Reviewer #1 (Remarks to the Author):

This is an interesting paper on the properties and role of the HMGA2 protein in neuronal progenitor cells. The authors use a large variety of techniques to this end and provide a large body of data supporting HMGA2 being involved in chromatin condensation and phase separation as well as being vital for neurogenesis. The data is compelling; however, there are parts where the data appears to be overinterpreted and raises limited concerns. Overall, this is a strong paper that needs some corrections/adjustments.

We thank the reviewer for his/her strong support on our study.

A broad range of biochemical approaches was used to explore the ability of HMGA2 to bind nucleosomes, condense polynucleosomes including AFM, form droplets in vitro and FRET based assays for folding linker DNA. They used His-tagged HMGA2 to show it binds to nucleosomes and does not directly bind to the linker histone H1.2. However, it is not clear if HMGA2 is directly binding the nucleosome or is merely binding the linker DNA present at both sides of the nucleosome and is merely being mediated by DNA. The authors should have performed these experiments with only the core nucleosome to be able to conclude direct binding to nucleosomes.

As suggested by the reviewer, we asked whether HMGA2 binds to nucleosome without linker DNA. We thus performed electrophoretic mobility shift assay with mononucleosomes wrapped with 145 bp DNA which does not include linker DNA. We found that HMGA2 binds to the core nucleosome without linker DNA (new Supplementary fig. 1b). The new results are now mentioned in our revised results (lines 146-148).

The polynucleosome experiments rely on the concentration of MgCl₂ required to aggregate polynucleosomes with or without HMGA2 (and H1.2) added. The biological relevance of this assay for measuring chromatin condensation is somewhat of a question in the field particularly in terms of the amounts of the varied amounts of MgCl₂ required for aggregation, so I'm not confident as to the over value of these experiments. The AFM experiments are probably a more preferred way for showing HMGA2 promotes condensation. From these experiments the extent of condensation seems to be modest and was wondering if the authors could make a more quantitative statement about the degree of change observed.

We agree that AFM experiments are a more direct and preferred way for showing chromatin

condensation. In the original manuscript, we examined the chromatin condensation activity of HMGA2 deletion mutants only with $MgCl_2$ -mediated condensation experiments. So, we newly performed AFM experiments with the series of HMGA2 deletion mutants (new Fig. 6c, d). According to the suggestion, we also added quantitative statement regarding the degree of change observed with the addition of HMGA2 in the AFM experiment (line 174).

The MNase experiments with mononucleosomes and HMGA2 and H1.2 were insightful as to the binding of these two proteins, but the statement that when both H1.2 and HMGA2 are present the extent of protection is greater than with H1.2 alone was not readily evident from the data provided. I don't think this needs to be the case for the point the authors are trying to establish.

We agree with the reviewer and removed the statement "when both H1.2 and HMGA2 are present the extent of protection is greater than with H1.2 alone" from the text.

The HMGA2 ChIP-seq done in parallel with ATAC-seq and DNase I-seq in mouse embryonic neocortical cells was very impactful and supports that HMGA2 binding causes DNA to be less accessible and likely more compacted in vivo. The finding from the Hi-C data that HMGA2 has a preference to be localized in B-compartments or transcriptionally inactive regions is further support for this point. The EGFP tagged Hmga2 locus mice provided strong evidence for Hmga2 being localized to condensed chromatin and together all of this provided compelling evidence for the role of HMGA2 in forming heterochromatic regions in vivo. The engineered NPC used to conditionally knock out Hmga1 and Hmga2 was a further demonstration of their role in gene repression and again is an excellent set of data. The droplet assay provides evidence for HMGA2 with nucleosomes promoting phase separation and the corresponding formation of droplets. More could be done with these assays, but in this current format is sufficient for this paper.

Thank you for the positive comments. We agree with the reviewer that more could be done with the droplet assay and added phase diagrams by observing the droplet in the presence of a range of varied HMGA2 protein and salt concentrations (new Fig. 4f). We found that HMGA2 formed droplets more efficiently at higher concentrations of HMGA2 and lower concentrations of salt, which is a typical feature of LLPS. These results further revealed the properties of HMGA2 in relation to LLPS and discussed in lines 239-242.

When performing domain deletions of HMGA2 to determine what are the domains required

for chromatin condensation, I was left with the question why the authors did not test the AT-hook 2 domain and instead only examined AT-hooks 1 and 3? The mutant HMGA2 proteins were only tested using the MgCl₂ aggregation assays which is probably an oversight, rather than the AFM assays for the reasons described above.

This is also an important point. We thus repeated the MgCl₂ dependent chromatin aggregation assay with a series of deletion mutants of recombinant HMGA2 protein including a hook2 deletion mutant and a hook3 deletion mutant. The result demonstrated that deletion of either hook1 or hook3 completely abolished the chromatin condensation activity of HMGA2 protein, while deletion of hook2 partially diminished it (new Fig. 6b, c). According to the suggestion, we also performed the AFM assays with the series of deletion mutants as mentioned before and found that deletion of either hook1, hook2 or hook3 abolished the chromatin condensation activity. These findings suggest that all AT-hook domains play a crucial role in chromatin condensation (lines 282-294).

The FRET assay used to assay the conformation of the linker DNA after binding of H1.2 and HMGA2 is potentially a powerful approach. I have not seen FRET data presented in this way and was looking for more quantitative analyses. The conclusion that HMGA2 restricts linker DNA movement is not clear or obvious given the extent of FRET change is not readily detected.

We agree with the reviewer and include the results with an improved experimental condition. It showed a significant (and obvious) increase of FRET signal by the addition of full-length HMGA2, suggesting that HMGA2 indeed restricts linker DNA movement. We conducted the assay using all the additional mutants we created (new Fig. 6g, h). The results revealed a correlation between the FRET signal and chromatin aggregation activity, providing further evidence that restricted linker flexibility may underlie chromatin condensation by HMGA2 (lines 310-311).

The other *in vivo* assays showing the AT-hook is important for puncta formation and *in vitro* for droplet formation were good and support the overall conclusions of the paper. The experiments to test whether the AT-hook 1 is required for HMGA2 maintenance of neurogenic progenitors were particularly important experiments. The bar graph showing the quantitative differences between wild type and AT-hook deletion mutant however did not seem to agree with the immunofluorescence images shown in Figure 7. When comparing full length versus AT-hook deletion the quantitation indicates nearly a 10-fold decrease, but I can't visually see

that in the image.

According to the suggestion, we replaced the images with more representative ones. Moreover, we also include Hoechst staining showing tissue architecture and integrity (new Fig. 7d, Supplementary Fig. 7).

Up until the end of the paper the message of HMGA2 has all been about gene repression, but at the end RNA-seq analysis is done after over expression of different forms of HMGA in the embryonic neocortex and the authors focus on the activation of transcription by HMGA2. I was left wondering how does HMGA2 activate transcription, how does this relate to the rest of the paper and why is the activation properties of HMGA2 the key for maintenance of proliferating cells rather than repression and chromatin condensation.

We actually do not know which of the activation or repression properties of HMGA2 are more important for maintenance of proliferating cells. What we found was that the hook1 domain is necessary for both activation and repression properties of HMGA2 (Fig. 7g-i) as well as for maintenance of proliferating cells (Fig. 7d, e). Our new results regarding categorization of HMGA2 bound genes now showed that the repression takes place mostly on the genes which harbor HMGA2 at the gene body (cluster 2) while the activation takes place in other clusters (e.g. *Igf2bp2* and *Plag1*, well-known direct HMGA2-activated targets, belong to cluster 5) (Fig. 5a-g). Gene activation by chromatin condensation can take place directly (e.g. *Igf2bp2* and *Plag1*) or indirectly (e.g. cell cycle genes shown in Fig. 7g). Regarding what mechanism might explain “direct gene activation by chromatin condensation”, we discuss in lines 484-486 that “Chromatin condensation is often associated with gene repression, but it may also promote gene activation such as by facilitating enhancer-promoter interactions in the case of LLPS induced by the Mediator complex”. This is a very important point to be addressed in future studies.

Reviewer #2 (Remarks to the Author):

The study by Kuwayama et al. investigates the mechanism of HMGA2-mediated regulation of gene expression. The authors identify histone H1 as one of the interaction partners of HMGA2 in the developing mouse neocortex, and, using in vitro reconstitution assays, show that HMGA2 promotes chromatin condensation. Motivated by this finding, they analyze HMGA2 genomic localization in the neocortex by ChIP-seq and correlate binding with open

chromatin (ATAC-seq, DNase I-seq) and A/B chromatin compartments (HiC). This analysis revealed reduced chromatin accessibility and preferential enrichment in B compartments, indicative of heterochromatin, for HMGA-bound sites. Based on in vitro data, the authors propose that HMGA2 has intrinsically disordered domains and forms droplets. Loss of HMGA1/2 or HMGA2 in neural progenitor cells lead to upregulation of HMGA2 bound genes. Using structural deletion mutants of HMGA2, AT-hook 1 is shown to be important for chromatin condensation. Finally, overexpression of HMGA2 in the developing mouse neocortex resulted in an increased number of progenitor cells, along with expression of neurogenic genes and reduction of astrogenic genes, which requires the AT-hook 1 or other unstructured domains to be present in HMGA2.

This work represents an elegant combination of in vitro biochemical assays and in vivo work in the developing neocortex, where HMGA2 contributes to gene regulation. Understanding the molecular mechanisms of gene regulation is important in the context of development and disease. In this manuscript, the authors analyze intrinsic protein properties and functions in vitro, and then move on to examine the relevance of their findings in a biological system of great relevance, the developing brain. Overall, this study greatly advances our understanding of the function of abundant non-histone chromatin proteins.

While the experiments are overall well-controlled and support the conclusions put forward in the manuscript, there are some points that should be considered:

We thank the reviewer for his/her strong support on our study.

1. The chromatin data (ChIP-seq, ATAC-seq, DNase I-seq, HiC) in Figure 3 requires more in-depth analysis to allow assessment of the quality of the data (replicates, location of peaks in relation to genomic features, similarity of ChIP with the different HMGA2 antibodies). Example tracks and peaks should be presented for representative loci. What is the nature of the repressive chromatin (i.e., genes, centromeres, telomeres, repeats etc.)? Is there any correlation with other repressive modifications (H3K27me3, H3K9me3)? While accession numbers to deposited data sets were included in the manuscript, the data was not accessible to the reviewer.

According to this great suggestion, we included more in-depth analyses of ChIP-seq data.

We performed HMGA2 ChIP-seq of four replicates each using two different antibodies. The correlation between them is more than 0.8, which suggests the high

reproducibility of the results (new Supplementary Fig. 2c).

We analyzed the location of the peaks and found that the proportion of the HMGA2 peaks in promoter regions was lower compared to the whole genome (new Supplementary Fig. 3a). Given that the promoter tends to be in an open chromatin state, this result is consistent with our observation that HMGA2 is enriched in condensed chromatin. We also analyzed the repeat sequences and found that HMGA2 signals were enriched at repeat sequences, especially at the satellite repeats compared to IgG control and HP1 signals, suggesting that HMGA2 is at least partially localized to the centromere, pericentromere and telomere (new Supplementary Fig. 3b,c).

By analyzing public data on the correlation with other repressive markers/factors, we found that HMGA2 has a low correlation with H3K27me3 and a high correlation with HP1, suggesting that HMGA2-bound genomic loci are different from H3K27me3-bound loci and overlap with HP1-bound loci (new Fig. 3e, new Supplementary Fig. 2b, h, i). Consistent with this observation, immunohistochemical analysis further showed that HMGA2 mostly colocalized with HP1 and H3K9me3 (new Supplementary Fig. 3j).

As suggested, we also added example tracks for chromosome 1 (new Fig.2b). HMGA2 signals were enriched in B compartment and roughly correlated with HP1, but tended to inversely correlate with DNase-seq, ATAC-seq and H3K27me3 signals. These results all suggest that HMGA2 localizes to condensed heterochromatin in neural progenitors of the developing neocortex.

These results are discussed in lines 208-212 and 223-225.

Regarding making datasets available, we made token for reviewers. The data will become open to anyone upon publication of this paper.

Sample name	GEA	Token
Hi-C_E11	E-GEAD-571	vqvGnyktWzHyP0KX2YxZ
RNA-seq_Hmga2_sKO	E-GEAD-572	qODsnOFhbCEc6CKkBtpl
RNA-seq_Hmga2_dKO	E-GEAD-573	dLxe35laJjESmvoV55TD
RNA-seq_Hmga2_OE 1	E-GEAD-574	H1Phf68rY2vQxnGV7e5m
HMGA2 ChIP-seq 1	E-GEAD-575	gGZxAO33Nm6ysv84VXtW
HMGA2 ChIP-seq 2	E-GEAD-576	7idSzKDZGAR6rjxAD03X
HMGA2 ChIP-seq 3	E-GEAD-577	GUyIYvDBCQ0ruMPDEh3K
HMGA2 ChIP-seq 4	E-GEAD-578	mGULobFbGkgCXuBmHUyC
ATAC-seq_E11	E-GEAD-581	UKHULZlhDA6ohx6aJxST
DNase-seq_E12	E-GEAD-582	F5ub0f8UA1MZ67oViel8

RNA-seq_Hmga2_OE 2	E-GEAD-624	IMAKUnIG29zmcEdBSP7A
------------	----------------------

2. In Figure 3F, a DNA stain would be helpful to appreciate the enrichment in the VZ. Are the cells on the right from tissue or cultured cells? If indeed VZ, it is surprising to see the low packaging of cells.

According to the suggestion, we included a DNA stain in the results (Fig. 3g). The cells on the right are from tissues. We changed the images to more representative ones (Fig. 3g). We also included the staining of heterochromatin factors/marks as discussed above (new Supplementary Fig. 3j).

3. Does droplet formation dependent on protein concentration (Figure 4)? Is this in a similar range to other proteins shown to undergo LLPS?

Yes, droplet formation by HMGA2 was dependent on its protein concentration (Fig. 6l, new Fig. 4f). We found that HMGA2 underwent LLPS at $> 5\sim 10 \mu\text{M}$, which is in a similar range of other proteins shown to undergo LLPS, such as MeCP2-GFP ($>2\sim 5 \mu\text{M}^1$) and HP1 ($>100 \mu\text{M}^2$). These results are discussed in lines 239-242.

4. For the RNA-seq expression analysis, it would be helpful to see the data presented as expression changes in HMGA2-bound regions. What fraction goes up/down? Which kind of genes?

We agree that this is a very important point. We therefore analyzed the gene expression changes in HMGA2-bound genes. First, we performed clustering of genes based on the HMGA2 binding pattern and found that some genes have higher amount of HMGA2 at their promoters (cluster1 and cluster3) and others have higher amount of HMGA2 at their gene bodies (cluster2) (new Fig. 5f). We found that Hmga1/2 knock out showed upregulation of genes in cluster2 on which HMGA2 is enriched in their gene bodies (new Fig. 5g). This suggests that HMGA2 is associated with gene repression when it binds to the gene body. This trend was also observed in Hmga2 single KO (new Supplementary Fig.5). We further confirmed these results by ChIP-seq with another HMGA2 antibody. Moreover, we found that cluster2 genes showed a higher level of expression compared with genes of other clusters (new Fig. 5h). These results indicated that binding of HMGA2 to the gene body of highly expressed genes is associated with transcriptional repression in embryonic neocortical NPCs. Furthermore, GO analysis showed that genes upregulated by Hmga KO were enriched in

genes related to neural development, implying that regulation of these genes are important for regulation of neural progenitor cell fate (new Fig. 5i). These points are discussed in lines 266-277.

6. In Figure 6F, is pericentric heterochromatin overall disturbed by expression of HMGA2 hook 1 del? In the DNA stain, the heterochromatin foci are not apparent.

We apologize for the confusion. In this assay, we used human-derived cells (IMR90) that originally do not exhibit clear heterochromatin foci. Consistent with previous reports, forced expression of HMGA2-GFP leads to the formation of aggregates known as SAHF (senescence-associated heterochromatin foci)³. Interestingly, we observed that the absence of hook1 prevents the formation of these aggregates compared to the full-length condition. This suggests that hook1 is crucial for the formation of aggregates by HMGA2-GFP. We newly included the data showing the control in which only GFP was expressed and increased the number of cells included in the representative image (new Fig. 6i, j).

7. For the in utero electroporation images in Figure 7, a DNA stain should be presented for all conditions to appreciate the effect on tissue architecture and integrity.

Thank you for the advice. We now added Hoechst staining in all in utero electroporation data to show that in utero electroporation did not cause apparent damage on tissue architecture and integrity (Fig 7d, Supplementary Fig. 7).

8. The authors propose that lack of the AT hook 1 in HMGA2 only affects the chromatin condensation function of HMGA2. However, given that AT hook domains in general are known to contribute to DNA binding, does the hook 1 deletion protein actually bind to chromatin in a similar way as the full-length HMGA2 protein? This should be tested by ChIP. If the hook 1 deletion protein does not bind to chromatin, this would be an alternative explanation for its reduced ability to induce HMGA2 phenotypes.

We agree that this is an important point. We therefore performed ChIP with HMGA2 antibody on electroporated samples. We found that the amount of immunoprecipitated DNA was comparable between full-length HMGA2 and hook1 del HMGA2 mutant (new Supplementary Fig. 6a). We also showed that hook1 del HMGA2 mutant binds to nucleosomes to the same extent with full-length HMGA2 in vitro (Fig. 6f). These results suggest that the effect of hook1 deletion is not simply due to reduced nucleosome binding ability. These points are discussed

in lines 296-302.

9. There is a typo in line 443 (McCP2).

We thank the reviewer for catching this typo and correct it.

Reviewer #3 (Remarks to the Author):

In this work, the authors studied 1) the interaction between HMGA2, H1 and nucleosomes; 2) the chromatin location and transcription regulation of HMGA2; 3) LLPS of HMGA2 and 4) HMGA2 maintaining neurogenic progenitors. They found some interesting results, including both HMGA2 and H1 interact with nucleosomes, HMGA2 localized to transposase 5- and DNase I- inaccessible chromatin regions, HMGA2 binding was mostly associated with gene repression, the AT-hook 1 domain was necessary for chromatin condensation by HMGA2 in vitro and in cellulo, and an HMGA2 mutant lacking this domain was defective in the ability to maintain neuronal progenitors in vivo. However, the relevance between these results has not been fully studied. I also have some concerns about some conclusions and experiment designations.

We thank the reviewer for constructive criticisms, which we address in detail below.

1. The first section title “HMGA2 interacts with linker histone H1” does not have enough supports in this work. The fact is that both HMGA2 and H1 interact with nucleosomes, which can explain all results in Fig. 1. The authors did not analyze the competition between HMGA2 and H1 by quantitative experiments, so it’s improper to explain the result with the statements that HMGA2 does not compete, but rather forms a complex, with H1 in cells (Page6 line140-141).

We thank the reviewer for raising this important point. We agree that our results do not show that HMGA2 does not compete with H1, and thus removed the corresponding sentences. Moreover, the statement “HMGA2 interacts with linker histone H1” may be misleading. So we changed the expression into “HMGA2 forms a complex with linker histone H1” based on the results of co-IP between HMGA2 and H1 (lines 116, 1159).

2. The authors analyzed HMGA2 ChIP-seq with ATAC-seq, DNase I-seq and AB

compartments, draw the conclusion that HMGA2 localizes to heterochromatin in the mouse neocortex. I think it's better to use more direct heterochromatin marker like ChIP-seq of HP1 and H3K9me3.

According to the suggestion, we analyzed HP1 ChIP-seq data from the neocortex of E13 embryo (we could not obtain reliable H3K9me3 data) and found that HP1 level is higher at HMGA2-bound regions compared to surrounding regions (new Fig. 3b and 3e and new Supplementary Fig. 2h). We also performed HP1 and H3K9me3 staining and compared them with endogenous HMGA2-GFP (new Supplementary Fig. 2j). We found that HMGA2 signals mostly overlapped with HP1 and H3K9me3 signals. These data further support the notion that HMGA2 localizes to heterochromatin in the mouse neocortex. These new data are now mentioned in our revised results (lines 208-212 and 223-225).

3. HMGA2 has three AT-hook domains with similar sequences, why did the authors only study AT-hook 1? Is it possible that three domains have same effects?

According to the suggestion, we made AT-hook2 del mutant and AT-hook3 single del mutant and performed the $MgCl_2$ aggregation and AFM assays (New Fig. 6b-e, New Supplementary Fig.10). The results demonstrated that deletion of either hook1 or hook3 completely abolished the chromatin condensation activity of HMGA2, while deletion of hook2 partially diminished it. These findings suggest that all hook domains play a crucial role in chromatin condensation, with hook1 and hook3 making particularly significant contributions to its activity (lines 283-294).

4. PUB1_{IDR}-HMGA2 hook1 del and HMGA2 hook1 del-FUS_{IDR} phenocopied full-length HMGA2. Can PUB1_{IDR}-HMGA2 hook1 del and HMGA2 hook1 del-FUS_{IDR} undergo LLPS in vitro with or without nucleosomes? Can they regulate the genes regulated by WT HMGA2?

These are also important points. We actually eliminated the data on HMGA2 hook1 del-FUS_{IDR} and focused just on PUB1_{IDR}-HMGA2 hook1 del, since we noticed that overexpression of HMGA2 hook1 del-FUS_{IDR} in neocortical NPCs sometimes resulted in cell toxicity (which we did not observe with PUB1_{IDR}-HMGA2 hook1 del). We purified the protein of PUB1_{IDR}-HMGA2 hook1 del and performed a droplet assay. We found that the PUB1_{IDR}-HMGA2 hook1 deletion protein forms droplets more efficiently in the presence of nucleosomes than in the absence of nucleosomes (new Fig. 7b, c). This result supports the notion that this IDR fusion rescued the effect of hook1 deletion on HMGA2-mediated

chromatin condensation at least in part.

We also analyzed the effect of PUB1_{IDR}-HMGA2 hook1 del on global gene expression profile. We indeed found that the suppression of HMGA2-dependent changes in global gene expression pattern by hook1 domain deletion tended to be rescued by the addition of PUB1_{IDR} to the HMGA2 hook1 del mutant at least in part (New Fig. 7i). This global tendency suggests that PUB1_{IDR}-HMGA2 hook1 del mutant can regulate the genes regulated by full length HMGA2. These new data are now mentioned in our revised results (lines 357-358 and 403-404).

5. HMGA2 cannot form LLPS droplet by itself, but PUB1IDR or FUSIDR undergo LLPS in vitro by themselves. Besides, PUB1IDR or FUSIDR are much longer than hook1 domain. I think such replacement experiments are not reasonable.

It is true that there are limitations in rescuing phenotypes through ectopic sequences. However, as mentioned above, the fusion of PUB1 IDR sequence to HMGA2 hook1 deletion mutant rescued the droplet forming ability in vitro and the neural progenitor maintenance ability in vivo, indicating a correlation between LLPS capacity and in vivo phenotypes. Moreover, the PUB1 IDR sequence is relatively short and suitable for ectopic addition experiments. Indeed, similar experiments have been conducted in several papers in which PUB1 IDR has been used⁴⁻⁶. Together, we think that these replacement experiments support the notion that droplet-forming activity is crucial for neural progenitor cell fate regulation via HMGA2.

6. Page6 line148: missing left bracket.

We thank the reviewer for pointing it out and correct it.

7. In the chart of Fig. 1A. Anti-HMGA2-#1 perform consistently with known findings, as well as the conclusions made here that HMGA2 is associated with H1 and H2B. While, it's confusing why the PSM MDC1 is "0" for the case of anti-HMGA2-#2, especially leading to inconsistent decisions about associations with H2B and MDC1. It's just slightly higher of the PSM of H1 proteins for anti-HMGA2-#2. Comprehensively compared with all these interacting proteins listed in 1A that were IP-ed using the two antibodies, anti-HMGA2-#2 performed worse.

We removed the IP-MS data obtained with HMGA2 antibody #2 as it did not efficiently

immunoprecipitate interacting proteins compared to HMGA2 antibody #1.

8. The authors should carefully arrange the different panels of Fig. 3f, and legend as well. For example, (f) Fluorescence microscopy of a coronal section of the Hmga2-EGFP mouse neocortex at E11.5 (top, scale bar = 200 μm) as well as of a portion of the ventricular zone also showing Hoechst 33342 staining (bottom, scale bars = 5 μm). What is the top or bottom panel referred to?

We apologize for the confusing format of original Fig. 3f and its legend. We reorganized the figure and changed the legend accordingly (lines 1228-1230).

9. Legends of fig.4a and 4b are lack of scalebars and indicated time for individual image. The line graph of FRAP is described that "Data are means \pm s.d. (n = 72 cells from three independent experiments)". How many droplets and cells are calculated here?

We now include scalebars and indicate time for individual images. In the FRAP assay, we measured fluorescence of one focus from one cell. Therefore, 72 droplets from 72 cells were calculated. We clarified this in the legend (lines 1238-1239).

10. Fig.4e and 4f. Actually, the optic resolution is far away qualified. Meanwell, the results of independent experiments should be supplied in sFigure. And also lack scalebar onto images. LLPS should be confirmed with a series of detail assays. It is very advised to construct phase diagrams by observing the droplet in the presence of a range of environmental conditions, typically varied protein concentrations and salt contents. The two phases start to separate in a switch-like manner. One should figure out the boundary.

Thank you for pointing out this error, which was caused by PDF conversion. We now fixed it. The results from independent experiments are also shown in Supplementary figure 10. We also added scalebars onto the image (Fig. 4g, h). According to the suggestion, we added phase diagrams with a range of protein and salt concentrations (new Fig. 4f). We found the droplet forming ability increased with increasing amounts of HMGA2 protein and decreasing concentrations of salt, which is a typical feature found in LLPS. We also found a switch-like droplet formation in a manner dependent on HMGA2 protein concentration (Fig. 6i). We discuss this point in the manuscript (lines 239-242).

11. Check order numbers in the legend of Fig. 6: e,f,g,f,g,i?

We rearranged the figures. Thank you for pointing this out.

12. Fig. 7: What's the plasmid introduced or expressed in control neocortex? Only GFP expressing plasmid?

Yes, the control plasmid expresses GFP alone. We clarified this in the method part (line 1067).

References

1. Li, C. H. *et al.* MeCP2 links heterochromatin condensates and neurodevelopmental disease. *Nature* **586**, 440–444 (2020).
2. Larson, A. G. *et al.* Liquid droplet formation by HP1 α suggests a role for phase separation in heterochromatin. *Nature* **547**, 236–240 (2017).
3. Narita, M. *et al.* A novel role for high-mobility group a proteins in cellular senescence and heterochromatin formation. *Cell* **126**, 503–14 (2006).
4. Li, W. *et al.* Biophysical properties of AKAP95 protein condensates regulate splicing and tumorigenesis. *Nat Cell Biol* **22**, 960–972 (2020).
5. Song, X. *et al.* Phase separation of EB1 guides microtubule plus-end dynamics. *Nat Cell Biol* **25**, 79–91 (2023).
6. Protter, D. S. W. *et al.* Intrinsically Disordered Regions Can Contribute Promiscuous Interactions to RNP Granule Assembly. *Cell Rep* **22**, 1401–1412 (2018).

REVIEWERS' COMMENTS

Reviewer #1 (Remarks to the Author):

The authors have addressed well my questions raised earlier and the revised manuscript is significantly improved. I recommend publication of the current version without any reservation.

Reviewer #2 (Remarks to the Author):

In the revised version of the manuscript, the authors have addressed all open points to my satisfaction. In particular, the authors have included additional analysis of the HMGA2 ChIP-seq and RNA-seq data, further characterizing the regions of the genome that are bound and regulated by HMGA2. Importantly, in the revised version, additional experiments were added to distinguish the effect of HMGA2 AT hook 1 deletion on chromatin binding vs. chromatin condensation. Moreover, a range of different in vitro experiments were added to further characterize the binding of HMGA2 to nucleosomes and the LLPS properties of HMGA2. Overall, this is an excellent study linking the in vitro biochemical characterization of HMGA2 to its functional role in neural fate regulation during in vivo neocortex development. I fully recommend this manuscript for publication.

Reviewer #3 (Remarks to the Author):

The Authors have addressed all of my concerns.